# Stability-Aware Feature Design for Robust Watermark Detection in Machine-Generated Text

Sina Mansouri [* 1]   Mohit Marvania [* 1]   Abolfazl Safikhani [2]

## Abstract

The widespread adoption of large language models (LLMs) has intensified the demand for principled methods to distinguish human- from machine-generated text. Watermarking provides a promising avenue, yet existing detectors exhibit sharp performance deterioration under multiple paraphrasing and when applied to shorter texts. We introduce *Pattern Stability Score (PSS)*, a novel detection framework that leverages local statistical features and stability dynamics across paraphrased variants. Specifically, the proposed method combines global and local z-score features with higher-order statistics of run-length patterns, enriched by autocorrelation signals and stability scores computed over paraphrase depth. Numerical evaluations are performed on three benchmark datasets (PG-19, CNN/DailyMail, and Wiki-Text) using multiple LLMs (Llama-3-8B, Qwen2-7B) and paraphrasers (Mistral-7B, Qwen2-7B, Gemma-7B), systematically stress-testing robustness under up to eight rounds of paraphrasing. Compared to prior z-score thresholding baselines and some state-of-the-art deep learning methods, our approach improves detection AUC (area under the receiver operating characteristic curve) by over 10-15 percentage points across different token lengths. Additionally, extensive cross-domain experiments demonstrate that a single universal classifier generalizes across different LLMs, paraphrasers, and text domains without retraining, maintaining above 87.8% AUC even when all components differ from training.

---
[*]Equal contribution [1]Department of Computer Science, George Mason University, Fairfax, VA, USA [2]Department of Statistics, George Mason University, Fairfax, VA, USA. Correspondence to: Abolfazl Safikhani <asafikha@gmu.edu>.

*Proceedings of the 43rd International Conference on Machine Learning*, Seoul, South Korea. PMLR 306, 2026. Copyright 2026 by the author(s).

## 1. Introduction

Large language models (LLMs) are now deployed at scale across both consumer and enterprise applications. As they increasingly integrate into writing workflows, the need to identify machine-generated content has shifted from a primarily academic inquiry to a practical requirement across domains such as education (Susnjak & McIntosh, 2024; Cotton et al., 2024), journalism (Chen & Shu, 2024; Zhou et al., 2023), and science policy (Blau et al., 2024; Gao et al., 2023), among others. A long history of *post-hoc* detection has been explored. For example, GLTR (Gehrmann et al., 2019) leverages rank histograms from a reference LM to highlight text that disproportionately employs high-probability tokens. This approach is efficient but its reliance on rank features renders it vulnerable to paraphrasing and domain variation. Similarly, Grover (Zellers et al., 2019) jointly trains a generator–discriminator pair, using an in-domain classifier for detection. However, the performance declines when either the generator or domain changes, with paraphrasing further diminishing robustness. More recent work, such as Binoculars (Hans et al., 2024), compares likelihoods under two open LMs and applies a likelihood-ratio style criterion for zero-shot detection. This improves cross-domain generalization but still exhibits sensitivity to paraphrasing and short inputs. Other methods, including curvature and rank-based tests such as DetectGPT and its variants (Mitchell et al., 2023), similarly rely on probability access from one or more LMs and remain susceptible to paraphrase smoothing.

In contrast, our focus is on *watermarking*, which offers several distinctive advantages: it enables a keyed hypothesis test with controllable false positive rates under the null, a level of statistical precision difficult to attain with learned post-hoc classifiers whose error rates shift across domains, requires only token identities on the detector side (removing dependence on proprietary probability distributions), and anchors attribution in the provider's secret key rather than in learned stylistic features. At a high level, watermarking operates as follows: the generator biases token selection toward a hidden "greenlist" so that downstream text exhibits detectable statistical structure, while remaining human-readable (Kirchenbauer et al., 2023; Qu et al., 2025; He et al., 2025; Lau et al., 2024). While alterna-

tive watermarking families have been proposed, including distortion-free schemes via inverse-transform sampling (Kuditipudi et al., 2024), exponential-sampling approaches (Aaronson, 2022), and cryptographically undetectable watermarks (Christ et al., 2024), our scope is the greenlist family since variants of it have been deployed at production scale (Dathathri et al., 2024). The standard detector aggregates evidence into a *global* z-score and compares it to a threshold. However, a determined adversary can paraphrase the text, diluting or locally rearranging this signal (Sadasivan et al., 2025; Cheng et al., 2025b; Mitchell et al., 2023; Bao et al., 2024). There exist several methods to modify the watermarking scheme to make it more robust with respect to certain adversarial attacks such as paraphrasing (see Section 2 for a review on other types of watermarking schemes). These studies motivate our design choice: instead of changing the generator to resist paraphrasing, we change the detector to exploit signals that paraphrasing preserves only imperfectly, namely local structure and stability across rewrites.

This detector-centric approach offers five practical advantages over modifying watermarking schemes. First, it provides *deployment compatibility*: simple greenlist schemes have already been adopted at production scale (Dathathri et al., 2024), and detector-only improvements require no changes to generation pipelines, model architectures, or serving infrastructure. Second, it enables *retroactive applicability*: enhanced detectors can identify watermarks in previously generated content without regeneration, providing immediate value for the vast corpus of already-watermarked text and for legal, educational, and journalistic settings where historical attribution matters. Third, it achieves *robustness through defense-in-depth*: treating the watermark as a fixed signal and extracting evidence through multiple statistical lenses (local, global, stability-based) avoids the new attack surfaces and scheme-specific patterns that adversaries can exploit in more complex schemes (Ren et al., 2024; Yu et al., 2025; Diaa et al., 2025). Fourth, it *preserves generation quality and efficiency* without the stronger biases or auxiliary models required by multi-bit and adaptive schemes (Xu et al., 2025; Feng et al., 2025). Finally, it approaches the *theoretical optimality limits* for fixed watermarking schemes through better statistical analysis (Li et al., 2025b), consistent with the principle of extracting all available information before declaring the need for stronger watermarks.

Recent works on watermarking repeatedly highlight two open gaps: robustness to *multi-step paraphrasing* and stability on *short texts* (Sadasivan et al., 2025; Cheng et al., 2025b). To address these gaps, we study a black-box adversary who can paraphrase any given text up to $K$ steps using a strong instruction-tuned LLM. The adversary does *not* know the watermark key or parameters and will preserve the

original semantics and approximate length. Let $x^{(0)}$ denote the original passage (possibly watermarked) and $x^{(k)}{}_{k=1}^{K}$ its paraphrases at depths $D1 \ldots DK$. The detector receives a single text at test time (any $x^{(k)}$ for $k = 0, 1, \ldots, K$). Our objective is to maintain detection power under paraphrasing and across different text lengths while controlling false positives on human-written content.

**Why global z-score fails under paraphrasing.** Greenlist watermarking (Kirchenbauer et al., 2023) induces a binary indicator sequence over tokens (green/non-green) and a corresponding z-score measuring deviation from the null. The binary indicator sequence is constructed as follows: for each token $s^{(t)}$ in the generated text, we assign a value of 1 if the token belongs to the green list $G^{(t)}$ and 0 if it belongs to the red list $R^{(t)}$. Specifically, at each position $t$, a hash function seeded by the previous token $s^{(t-1)}$ deterministically partitions the vocabulary into a green list of size $\gamma|V|$ and a red list of size $(1 - \gamma)|V|$, where $\gamma$ is typically 0.25 or 0.5. During watermarked generation, tokens from the green list are softly promoted by adding a bias $\delta$ to their logits. At detection time, we reconstruct this binary sequence by checking whether each observed token $s^{(t)}$ falls in its corresponding green list $G^{(t)}$ (assigned 1) or red list $R^{(t)}$ (assigned 0), using the same hash function and seed. Global tests summarize all tokens into one statistic (e.g. z-score) that depends only on the total green-token count, not on where green tokens appear in the sequence. Paraphrasing replaces some green tokens with non-green tokens, and crucially, this replacement is *spatially non-uniform*: some regions retain strong watermark signals while others are heavily edited. The global $z$-score averages over the full sequence, so concentrated evidence in surviving regions is diluted by the heavily-edited regions, reducing the aggregate statistic even when localized evidence remains strong. A local detector, in contrast, can recover this heterogeneous signal by identifying windows where watermark evidence concentrates, the core motivation for our approach. The failure is structural: a single aggregate discards *where* evidence concentrates and *how* it behaves under edits.

**Key idea: stability-aware local detection.** The proposed method *Pattern Stability Score (PSS)*, is a detection framework that (i) extracts *local* watermark evidence via a rolling window and (ii) quantifies *stability* of that evidence across paraphrase depth. Specifically, we slide a window along the 0-1 sequence and compute several local statistics within each window. Note that when a window splits a consecutive sequence/run of ones, we expand it minimally so runs are not fragmented while shrinking the tail window to cover all remaining tokens. For each window we compute a 20-dimensional feature set: six summary statistics of the z-score sequence across windows (mean, variance, min, max, skew, and kurtosis), lag-1 and lag-2 autocorrelations

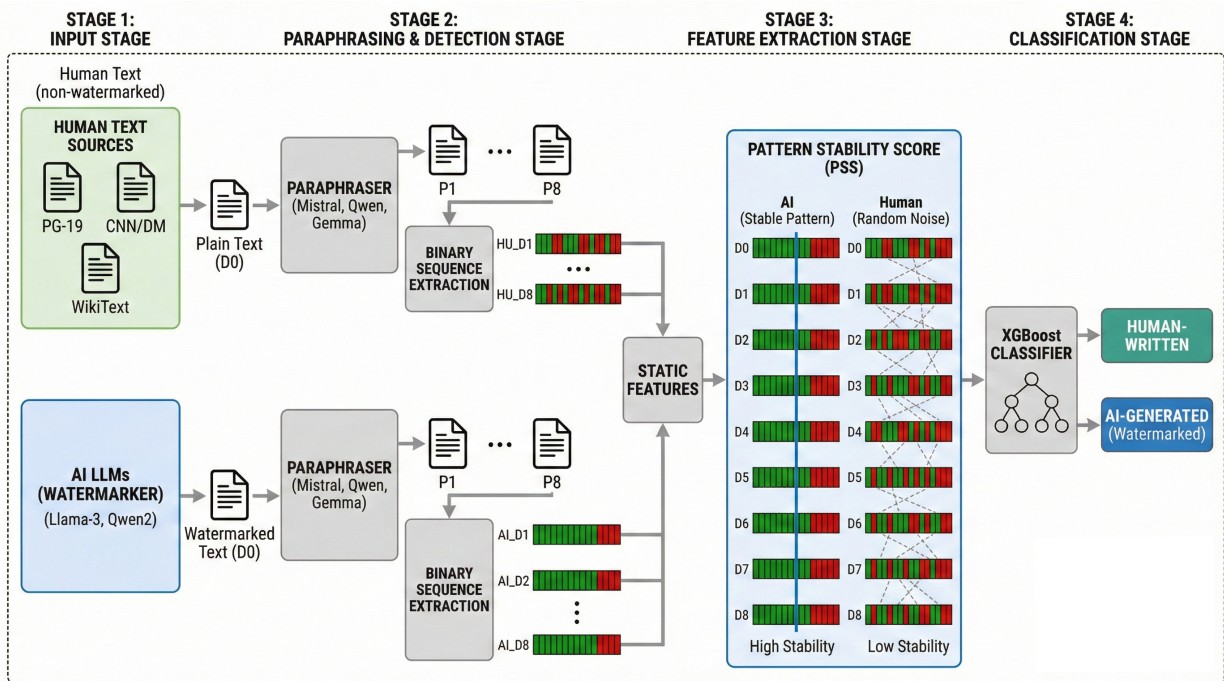

*Figure 1.* **End-to-end PSS pipeline.** Human and AI-watermarked texts undergo up to eight paraphrasing rounds, with binary sequences extracted at each iteration (D0–D8). Watermarked text maintains stable patterns across iterations while human text shows random variation. An XGBoost classifier uses these stability features alongside static features for final classification.

of z-scores, six summary statistics of longest-run length in the binary sequence as well as the same six summary statistics of frequency of the longest run. Aggregating these per-window features yields robust *local* statistics. We then compute a *pattern stability* functional, PSS over the trajectory $x^{(j)} \rightarrow \cdots \rightarrow x^{(K)}$, where Dj is the given text for some $j = 0, 1, \ldots, K-1$. PSS is computed by extracting per-window local z-scores across all participating paraphrased versions ($Dj$ to $DK$), aligning them to the minimum window count for consistency, and computing the standard deviation across depths. Specifically, for each window position $w_i$, we calculate $\text{PSS}_i = \text{std}(z_i^{(j)}, \ldots, z_i^{(K)})$ where $z_i^{(k)}$ denotes the local z-score at window $i$ for depth $k$ for $k = 0, 1, \ldots, K$ while std(.) denotes the standard deviation function. This window-wise variability signal is then concatenated with 20 static features to form the complete feature vector (see Section 3.4 for more details). Then, a simple classifier (e.g., XGBoost or logistic regression) on these hybrid features produces the final decision using a 70/30 stratified train/test split. The rationale behind the proposed detector is that to improve the detection power among potential multi-step paraphrasing, the method fuses two main ingredients: (i) *local* rolling-window statistics that preserve spatial structure of watermark evidence (moments of local z-score, short-range autocorrelations, longest-run length and frequency), and (ii) uncertainty metric (PSS)

that aggregates these features across paraphrase depths to capture both central tendency and variability (depth-wise variance and optional concordance). Local features expose pockets of concentrated green evidence that global tests average away, while PSS down-weights brittle depth-specific artifacts and rewards signals that persist under paraphrasing. This combination converts paraphrase-invariant regularities into separable features, yielding stable AUC (area under the receiver operating characteristic curve) at high depths and short lengths (see Section 4). Figure 1 visualizes the end-to-end pipeline, while Algorithm 1 in the Appendix provides the pseudocode.

The evaluation is conducted on a balanced corpus constructed from three benchmark datasets: PG-19 (long-form books) (Rae et al., 2020), CNN/DailyMail (news articles) (Hermann et al., 2015), and WikiText (Wikipedia content) (Merity et al., 2017). We generate watermarked passages using multiple LLMs including Llama-3-8B and Qwen2-7B with the greenlist watermarking method (Kirchenbauer et al., 2023) under configuration parameters $\gamma=0.25, 0.5$ (greenlist ratio), $\delta=1.5$ (bias), and a fixed hash key. To assess robustness, each passage is paraphrased for up to $K=9$ iterations (with evaluation performed on $D1$–$D8$; $D9$ enables PSS computation at $D8$) using three different paraphrasers (Mistral-7B-Instruct, Qwen2-7B-Instruct, and Gemma-7B-IT). We compare against both traditional baselines (global

z-score) and some state-of-the-art deep learning methods including DeepTextMark (Munyer et al., 2024), Binoculars (Hans et al., 2024), RADAR (Hu et al., 2023). Our approach substantially outperforms all baselines: at 1,500 tokens and depth $D8$, PSS + Static achieves 91.2% AUC while deep learning methods collapse to 41–44%.

Summary of main contributions are as follows: **(1) Stability-driven detection:** We introduce the PSS, a principled measure that captures the persistence of watermarking signals across successive paraphrasing depths. Beyond formalizing this stability perspective, we demonstrate how PSS can be effectively integrated with *local* rolling-window statistics to enhance detection granularity; **(2) Compact hybrid feature design:** We construct a 20-dimensional window-based feature set that incorporates statistical moments, autocorrelation descriptors, and run-length structural properties. This compact representation is deliberately engineered to maintain discriminative power even under aggressive paraphrasing and in short-text regimes, addressing key limitations of prior approaches; **(3) Robustness under adversarial stress:** Through systematic evaluation on three benchmark datasets (PG-19, CNN/DailyMail, WikiText), subjected to up to eight rounds of paraphrasing and reduced passage lengths as short as 300 tokens, we show that PSS consistently surpasses global z-score baselines and some state-of-the-art deep learning methods in AUC; **(4) Cross-domain generalization:** We demonstrate that a single universal classifier generalizes across different LLMs, paraphrasers, and text domains without retraining, addressing critical concerns about practical deployment where the attacker's configuration is unknown; **(5) Comprehensive sensitivity analysis:** We analyze the influence of critical hyperparameters, including window size, stride, and input length, on detection performance. The empirical results confirm that the proposed method remains robust under moderate parameter variations, reinforcing the reliability and practical deployability of PSS in diverse settings.

The rest of the paper is organized as follows. Section 2 reviews watermarking and detection methods while Section 3 formalizes proposed methods, namely local features and PSS. Section 4 details datasets, paraphrasing, metrics, and then presents empirical results. Finally, Section 5 covers some concluding remarks, limitations, and future research directions. The Appendix contains extended numerical analyses, sensitivity test details, and provided pseudocode.

## 2. Related Work

**Watermarking for LLMs.** Greenlist watermarking biases token sampling toward a partition of the vocabulary determined by a keyed hash. Detection then tests whether the realized proportion of "green" tokens is unusually high under the null (Kirchenbauer et al., 2023). The standard

detector reduces the problem to a single global z-score with a fixed threshold (often z-score $> 4$). Follow-up work characterizes trade-offs among bias strength, quality, and false positives, and analyzes limits under channel constraints and adversarial distortion (Qu et al., 2025; He et al., 2025; Lau et al., 2024). These approaches assume that compressing evidence into one statistic retains power; in practice, global aggregation is fragile when text is paraphrased or short.

A line of work investigates how paraphrasing and distribution shift erode detector power. Paraphrasing attacks—produced by instruction-tuned models or controlled editing—can disperse local green runs, alter token-level dependencies, and reduce the global statistic while preserving semantics (Cheng et al., 2025b; Sadasivan et al., 2025). Beyond watermark-specific detectors, post-hoc detectors such as DetectGPT and its accelerations exploit curvature or log-likelihood perturbations to separate human and model text, but they also degrade under paraphrases or domain shift (Mitchell et al., 2023; Bao et al., 2024). Our multiple rounds of paraphrasing follows this literature: a black-box paraphraser generates a depth-$K$ chain ($D1 \ldots DK$) without access to watermark keys, aiming to flip the detector while keeping meaning (Sadasivan et al., 2025; Cheng et al., 2025b; Rastogi & Pruthi, 2024). More targeted attacks include self-information rewrite attacks (Cheng et al., 2025a), which preferentially rewrite high-information tokens likely to carry watermark signal, and watermark-stealing attacks (Jovanović et al., 2024), which attempt to learn the greenlist partition without access to the key.

**Deep Learning-Based Detection Methods.** Recent works have explored deep learning approaches for detecting machine-generated text. For example, DeepTextMark (Munyer et al., 2024) employs neural networks trained on stylistic and statistical features of watermarked text while Binoculars (Hans et al., 2024) uses likelihood ratios from two open LMs for zero-shot detection. Also, RADAR (Hu et al., 2023) combines adversarial training with robust feature extraction, and commercial classifiers such as Pangram (Emi & Spero, 2024) push post-hoc detection accuracy further on in-distribution data. While these methods achieve high AUC on in-distribution data, our experiments (Section 4.3) reveal their catastrophic failure under paraphrasing attacks, with AUC dropping from 91–96% at D0 to 41–44% at D8. In contrast, our PSS-based approach maintains above 91% AUC even at D8, demonstrating that simple statistical features capturing watermark invariants significantly outperform complex deep learning architectures when robustness is required.

**Adaptive and Alternative Watermarking Schemes.** Adaptive schemes modify partitioning or biasing as generation proceeds, or modulate the watermark via content- or entropy-aware policies (Feng et al., 2024; Lau et al., 2024). Theo-

retical analyses characterize fundamental limits, e.g., how much capacity is available for reliable marking under a given distortion budget and adversarial rewrite power (He et al., 2025; Qu et al., 2025). Recent semantic approaches move beyond token-level manipulation, with SemaMark (Ren et al., 2024) introducing semantic embeddings for vocabulary partitioning rather than token hashes, providing robustness to paraphrasing attacks by maintaining semantic consistency. Similarly, semantic invariant watermarks (Liu et al., 2024a) generate watermark logits based on semantic context using embedding models, while SAEMark (Yu et al., 2025) employs Sparse Autoencoders to embed watermarks through feature-based rejection sampling on neural activations. Production-scale deployment has been achieved with SynthID-Text (Dathathri et al., 2024), which introduces tournament sampling with provable non-distortion properties and serves over 20 million responses in Google Gemini, validating the feasibility of pattern-based approaches at scale. Publicly-detectable watermarking (Fairoze et al., 2025) achieves distortion-free watermarking with cryptographic signatures via rejection sampling, incorporating error-correction for low-entropy periods. Recent work also targets dual-threat robustness, with detectors designed to resist both scrubbing (removal) and spoofing (forgery) attacks simultaneously (Shen et al., 2025). These studies motivate our design choice: instead of changing the *generator* to resist paraphrasing, we change the *detector* to exploit signals that paraphrasing preserves only imperfectly—namely local structure and stability across rewrites.

**Local vs. global statistics for detection.** Global tests ignore *where* evidence concentrates. Local analyses (rolling windows, run-length distributions, short-range autocorrelations) preserve spatial structure that is costlier for paraphrasers to randomize without semantic drift. An early example of detector-side local analysis is WinMax (Kirchenbauer et al., 2024), which scans for the single window with the highest local z-score and uses that maximum as the test statistic; this aggregates locality into a one-dimensional summary, whereas our approach summarizes the entire *distribution* of local statistics across windows and additionally measures their stability across paraphrase depths. Recent frequency-based approaches like FreqMark (Xu et al., 2024) employ Short-Time Fourier Transform for sentence-level detection with periodic signal embedding, achieving AUC up to 0.98 through windowing approaches that parallel our local detection strategy. Closely related detector-side work has tackled localization within mixed human/AI text (Zhao et al., 2025) and detection of post-generation edits via combinatorial watermarking (Xie et al., 2025), both of which exploit spatial structure in ways complementary to ours. Adaptive watermarking (Liu & Bu, 2024) uses entropy-based token selection with semantic logits scaling, selectively watermarking high-entropy distributions for improved robustness.

Statistical frameworks (Li et al., 2025b) provide closed-form expressions for asymptotic error rates and mathematically optimal detection rules, while likelihood-based detection (Li et al., 2025a) estimates null token probabilities for accurate detection, achieving approximately 65% power improvement over baselines. Universal optimality results (Li et al., 2025b) characterize minimum Type-II error for any watermarking scheme, establishing fundamental limits. Multi-bit approaches like MajorMark (Xu et al., 2025) implement clustering-based majority voting with block partitioning, while BiMark (Feng et al., 2025) achieves 30% higher extraction rates for short texts through multilayer architecture with bit-flip unbiased mechanisms. Ensemble watermarks (Niess & Kern, 2025) combine acrostic patterns, sensorimotor norms, and red-green watermarks, achieving satisfactory detection rate compared to red-green alone after paraphrasing. Linguistic-feature or style-based detectors (Gehrmann et al., 2019; Ding et al., 2019; Guo et al., 2024) implicitly leverage locality but are unkeyed and risk false positives on atypical human styles. Our method remains keyed to the watermark while augmenting the global test with compact local statistics. Empirically, this hybrid design–local moments and autocorrelations of z-score, longest-run and its frequency–closes much of the robustness gap under paraphrasing and short lengths, while keeping computation modest and features interpretable.

**Paraphrasing Detection and Inversion.** Orthogonal to watermarking, paraphrasing-detection methods attempt to identify machine paraphrase patterns directly, e.g., by modeling machine paraphrasing behavior or by inverting paraphrases (Krishna et al., 2023; Lu et al., 2025). Adaptive attacks using Direct Preference Optimization achieve over 96% evasion rate against surveyed watermarks (Diaa et al., 2025), while cross-lingual attacks (He et al., 2024) reveal fundamental weaknesses, with Cross-lingual Watermark Removal Attack decreasing AUCs from 0.95 to 0.67. Comprehensive evaluations (Liu et al., 2024b) show KGW achieving only 0.0349 watermark rate under paraphrase attacks, demonstrating the need for multi-attack robustness testing. Domain-specific challenges further complicate detection: SWEET (Selective WatErmarking via Entropy Thresholding) (Lee et al., 2024) addresses code's low entropy by watermarking only high-entropy segments, while medical text evaluation (Hastuti et al., 2025) shows current watermarking methods compromise medical factuality, introducing Factuality-Weighted Score metrics that prioritize accuracy over detectability. These approaches can complement watermark detectors but rely on assumptions about the paraphrasing model and break when attackers switch paraphrasers. Our setting treats the paraphraser as a black box, remaining agnostic to the model family and focusing on the behavior of watermark evidence under paraphrasing.

# 3. Proposed Methodology

In this section, we describe the proposed watermark detector, including *local* statistics, the *Pattern Stability Score* (PSS) computed across paraphrase depth with multiple paraphrasing, and the final classifier.

## 3.1. Preliminaries: Greenlist Watermarking and the Global Test

Let $\mathcal{V}$ be the vocabulary and let $h(\cdot; k)$ be a keyed hash that maps a token-context pair to $[0, 1]$. For a partition parameter $\gamma \in (0, 1)$, the *greenlist* at position $t$ is

$$G_t = \{v \in \mathcal{V} : h(v, x_{<t}; k) \leq \gamma\},$$

where $x_{<t} = (x_1, x_2, \ldots, x_{t-1})$ denotes the sequence of tokens preceding position $t$. During generation, the model increases the probability mass on $G_t$ by a bias $\delta > 0$. Given a token sequence $x_{1:n}$, define the indicator $b_t = \mathbf{1}\{x_t \in G_t\}$ and the global test statistic (i.e. z-score)

$$z(x_{1:n}) = \frac{\sum_{t=1}^{n} b_t - n\gamma}{\sqrt{n\gamma(1-\gamma)}}.$$

Classical detection declares "watermarked" if z-score $> \tau$ for a fixed threshold (often $\tau{=}4$). This test is efficient and interpretable but discards spatial information and is known to degrade under paraphrasing (Sadasivan et al., 2025).

## 3.2. Multi-Step Paraphrasing and Data Generation

We assume a black-box paraphraser that maps any text $x^{(0)}$ to a sequence of paraphrases $\{x^{(k)}\}_{k=1}^{K}$, preserving semantics and approximate length, without access to $k$ or $(\gamma, \delta)$. In our pipeline: (i) watermarked texts are generated with an open LLM using standard $(\gamma, \delta)$ and sampling; (ii) each text is paraphrased up to depth $K{=}9$; (iii) detection is evaluated on depths $D1$–$D8$ (with $D9$ used only for PSS computation at $D8$). Exact prompts and decoding settings are specified in Section 4. We evaluate multiple lengths, i.e. $n \in \{300, 500, 1000, 1500\}$ tokens.

## 3.3. Local Rolling-Window Statistics

Global aggregation ignores *where* watermark evidence concentrates. We therefore compute *local* features by sliding a window of size $w$ with stride $s$ across $x_{1:n}$ and its indicator sequence $b_{1:n}$ to compute local z-scores.[1] Let the $i$-th window cover indices $t \in [a_i, b_i]$. For each window we compute:

1. **Local z-score summary** over $\{b_t\}_{t=a_i}^{b_i}$ defined as

---

[1]We use $w{=}50$, $s{=}10$ by default. If a window boundary splits a consecutive run of ones in $b_{1:n}$, we expand the window minimally to keep the run intact; the tail window shrinks to cover remaining tokens. Sensitivity to $(w, s)$ is reported in the Appendix.

$z_i = \frac{\sum_{t=a_i}^{b_i} b_t - m_i\gamma}{\sqrt{m_i\gamma(1-\gamma)}}, \qquad m_i = b_i - a_i + 1$, and compute six summary statistics of $\{z_i\}$ sequence across windows: mean, variance, min, max, skew, kurtosis.

2. **Autocorrelation of local z-score** across windows: $\rho_z(\ell)$ at lags $\ell \in \{1, 2\}$.

3. **Run-length statistics** inside the window: the longest consecutive run of ones $R_i$, and its frequency $F_i$ (number of occurrences). We then compute the six summary statistics of $\{R_i\}$ and of $\{F_i\}$ across windows.

This yields a compact 20-dimensional *local feature set*: 8 from z-score (6 summary statistics + 2 autocorrelations), 6 from run-length, and 6 from run-frequency. These capture spatial concentration and short-range dependencies that paraphrasers disrupt only imperfectly without semantic drift.

## 3.4. PSS Across Paraphrase Depth

Paraphrasing aims to rearrange evidence. If the underlying text is watermarked, we expect local evidence to persist across mild rewrites; for human text, local evidence should fluctuate around the null. We formalize this intuition via a stability functional over the local z-score trajectories $\{x^{(k)}\}_{k=0}^{K}$. across paraphrase depths. Given a text at an unknown paraphrase depth $j \in 0, 1, \ldots, K$, we generate its subsequent paraphrases up to depth $K$ to obtain the sequence $\{x^{(k)}\}_{k=j}^{K}$. For evaluation at depth $Dj$, PSS is computed using depths $Dj$ through $DK$, ensuring at least two points for the standard deviation calculation. For each text in this sequence, we compute local z-scores using the rolling-window procedure described in Section 3.3. Because different paraphrase depths can yield different numbers of windows, we align all sequences to the minimum window count across depths before aggregation. The Pattern Stability Score is then computed by measuring the variability of local z-scores at each window position across depths. Specifically, for each aligned window position $i$, we calculate:

$$\text{PSS}_i = \text{std}(z_i^{(j)}, z_i^{(j+1)}, \ldots, z_i^{(K)}),$$

where $z_i^{(k)}$ denotes the local z-score at window position $i$ for paraphrase depth $k$. This computation yields a vector of stability scores, one for each window position, capturing how consistently the watermark signal manifests at each local region across paraphrasing transformations. The intuition behind PSS is that watermarked text shows more stable local patterns across paraphrases than human text. When a text is genuinely watermarked, the underlying statistical bias persists even as surface tokens change through paraphrasing, resulting in relatively consistent local z-scores and thus lower PSS values at each window position. Conversely, human text subjected to paraphrasing shows higher variability in local z-scores across depths, as there is no underlying

watermark signal to maintain consistency. The complete feature vector for classification consists of two components: (i) the PSS values computed across all window positions, providing a stability profile of the text, and (ii) the static features extracted from the current text, including summary statistics of z-scores, run-length patterns, and frequency statistics as defined in Section 3.3. This hybrid approach combines the temporal stability information from PSS with the instantaneous statistical patterns from static features.

## 3.5. Classifier and Decision Rule

Given a passage at an unknown paraphrase depth $Dj$, we (i) compute the greenlist indicator $b_{1:n}$ and local z-scores, (ii) extract the *local* rolling-window feature set from Section 3.3 (20-dimensional statistical values, short-range autocorrelations (lags 1 and 2), and run-length statistics), and (iii) optionally augment these with *stability* features via the PSS from Section 3.4, which aggregates depth-wise consistency of the same local statistics. The resulting feature vector $g(x)$ is fed to a lightweight supervised classifier that outputs a posterior $p_\theta(y{=}1 \mid g(x))$ and a binary decision via a fixed threshold. We compare four standard learners on $g(x)$, logistic regression, random forest, XGBoost, SVM (RBF), and $k$NN, chosen for complementary bias/variance profiles and interpretability. Classifier hyperparameters use library defaults unless noted. Unless stated, XGBoost serves as the representative classifier in numerical results (it consistently ranks first or tied in our empirical studies). For each depth and input-length regime, we use a stratified 70/30 train/test split with fixed random seed. Hyperparameters follow library defaults unless noted, no test-time tuning is performed. At inference, we report AUC, together with Precision/Recall/F1. Figure 1 summarizes the modules and data flow.

# 4. Experiments and Results

We evaluate robustness to paraphrasing and short texts on three benchmark datasets. This section specifies the data and multiple paraphrasing, the compared methods, the training/evaluation protocol, and the main numerical results including some robustness. Extended tables and plots are deferred to the Appendix.[2]

## 4.1. Experimental Setup

We evaluate on three diverse text corpora: (1) **PG-19** (Rae et al., 2020): 1,000 human-written books representing long-form literary content; (2) **CNN/DailyMail**: 1,000 news articles representing journalistic writing; and (3) **WikiText**: 1,000 Wikipedia passages representing encyclopedic con-

tent. For each dataset, we generate an equal number of watermarked passages using multiple LLMs including Llama-3-8B and Qwen2-7B. Each passage is truncated to a fixed token length. We report four length regimes: 300, 500, 1,000, and 1,500 tokens. We use standard greenlist watermarking with partition parameter $\gamma{=}0.25$ (and $\gamma{=}0.5$ for additional experiments), bias $\delta{=}1.5$, and a fixed hash key. Decoding uses nucleus sampling and moderate temperature, the detector uses the same $(\gamma, \delta)$ and hashing as the generator. We evaluate three paraphrasers: Mistral-7B-Instruct (primary), Qwen2-7B-Instruct, and Gemma-7B-IT, each rewriting passages up to depth $K{=}9$ under a length-preserving prompt. We denote the original as $D0$ and the $k$-th paraphrase as $Dk$. Evaluation is performed on depths $D1$–$D8$; depth $D9$ is generated solely to enable PSS computation at $D8$ (since PSS requires at least one subsequent depth). At test time the detector receives *one* text (potentially any $Dk$ for $k \in \{1, \ldots, 8\}$) without access to other depths.

## 4.2. Methods Compared

We compare eight detectors that differ only in feature design, all use the same training protocol. (1) **Global z-score threshold:** The canonical one-sided test declares "AI" if z-score $\geq \tau$ with $\tau{=}4$. (2) **Local z-scores (20-D):** We compute rolling-window local $z_i$ across the passage and then form a *fixed-length vector* from the first 20 window $z_i$'s (if more windows exist, we uniformly subsample to 20; and we make sure the included sequence have at least 20 z-scores). This 20-dimensional (20-D) *raw local pattern* is fed to LR/RF/XGB/SVM/KNN classifier models. (3) **Static features (20-D):** Instead of raw $z_i$ values, we summarize *across windows* with compact statistics that preserve locality and reduce dimensionality: for $\{z_i\}$, six moments (mean/var/min/max/skew/kurtosis) plus lags-1 and 2 autocorrelations (8 features), for the longest run length per window $\{R_i\}$, six moments (6 features); and for the frequency of the longest run per window $\{F_i\}$, six moments (6 features). In total, there are 20 features which are provided to the same classifier models. (4) **PSS + static features:** We compute PSS values (see Section 3.4) over paraphrase depth using the same local statistics to quantify depth-wise consistency. Then, we concatenate PSS with a selected subset of the static features and train an XGBoost classifier (our best single model). (5) **DeepTextMark** (Munyer et al., 2024): A neural network-based detector trained on stylistic and statistical features of watermarked text. (6) **Binoculars** (Hans et al., 2024): Zero-shot detection using likelihood ratios from two open language models. (7) **RADAR** (Hu et al., 2023): Adversarially trained detector with robust feature extraction for machine-generated text detection. (8) **WinMax** (Kirchenbauer et al., 2024): Local detector for greenlist watermarking that scans the binary indicator sequence with a fixed-width window and reports the *maximum* local z-score

---

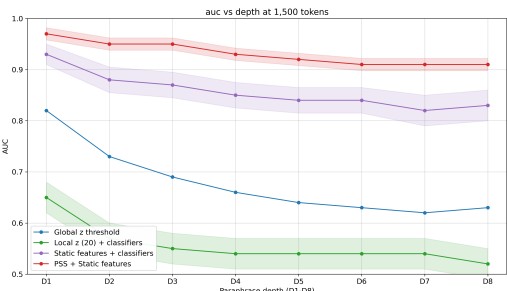

*Figure 2.* **AUC vs. paraphrase depth at 1,500 tokens.** Mean AUC (solid lines) with $\pm 1$ SD bands (shaded) over 30 runs with random 70/30 splits for four methods—Global z threshold, Local z (20) + classifiers, Static features + classifiers, and PSS + Static—all using XGBoost. Local statistics improve robustness relative to the global threshold, and adding PSS further flattens the AUC decline from $D3$–$D8$.

as the test statistic, providing a one-dimensional summary of the strongest local watermark concentration.

### 4.3. Empirical Results

**Performance at 1,500 tokens.** Figure 2 depicts AUC against paraphrase depth $D1$–$D8$. The global z-score threshold is competitive at shallow depth but declines steadily as paraphrasing deepens. Injecting locality slows this drop: both *local z (20-D) + classifiers* and static features + classifiers yield flatter AUC curves than the global statistic. The largest gains come from adding *stability*: *PSS + Static* remains comparatively flat through mid/late depths, indicating that cross-depth persistence provides signal beyond any single local snapshot. To unpack where this signal comes from, we incrementally add components to the global z-score baseline (full breakdown in Table 6 in the Appendix). Z-score moments (6-D) carry the strongest individual contribution, lifting the global baseline by 5–9 percentage points across depths and establishing the foundation for local detection. Adding lag-1/lag-2 autocorrelations (8-D) provides further gains by capturing short-range spatial dependencies that distinguish coherent watermarked structure from near-zero autocorrelation in human text. Run-length statistics (14-D) capture the structural signature of greenlist biasing, with their contribution becoming more pronounced at deeper paraphrase depths where the z-score signal alone weakens. The full 20-D static feature set yields a 12–14 percentage point improvement over the global baseline at every depth, and PSS provides the largest single jump (8–13 additional percentage points) by adding the cross-depth stability dimension. Each feature group addresses a distinct, non-redundant aspect of the watermark signal. To ensure consistent evaluation across all depths, we generate paraphrases up to $D9$, enabling PSS computation at $D8$ using the stability between $D8$ and $D9$.

*Table 1.* **Comparison with prior detectors on PG-19 at 1,500 tokens.** AUC (%) across paraphrase depths. PSS + Static outperforms statistical baselines (Global z-score, WinMax) by 14–19 percentage points an, the global z-score by 22–24 percentage points, and deep-learning detectors (DeepTextMark, Binoculars, RADAR) by 30–50 percentage points across $D1$–$D8$.

| Method | D1 | D3 | D5 | D8 |
|---|---|---|---|---|
| Global z-score | 74.2 | 70.0 | 68.0 | 66.8 |
| WinMax (Kirchenbauer et al., 2024) | 82.1 | 76.5 | 74.4 | 72.3 |
| Static features (20-D) | 88.1 | 83.2 | 80.2 | 78.6 |
| DeepTextMark (Munyer et al., 2024) | 65.8 | 51.2 | 47.5 | 43.9 |
| Binoculars (Hans et al., 2024) | 62.7 | 48.6 | 44.9 | 41.9 |
| RADAR (Hu et al., 2023) | 58.4 | 47.2 | 43.7 | 40.8 |
| **PSS + Static** | **96.1** | **93.9** | **92.6** | **91.2** |

**Comparison with prior detectors.** We compare PSS + Static against statistical baselines (global z-score, WinMax (Kirchenbauer et al., 2024)) and deep-learning detectors (DeepTextMark (Munyer et al., 2024), Binoculars (Hans et al., 2024), RADAR (Hu et al., 2023)) on PG-19 at 1,500 tokens (Table 1). WinMax improves on the global baseline by exploiting locality (82.1% at $D1$), but reducing all local information to a single maximum value leaves a 14–19 percentage point gap to PSS + Static. The deep-learning detectors, while competitive at $D0$ (91–96% AUC), collapse catastrophically under paraphrasing, dropping to 41–44% AUC at $D8$ as learned features overfit to surface statistics that paraphrasing destroys. PSS + Static maintains 91–96% AUC across all depths, showing that compact statistical features capturing watermark invariants outperform complex neural architectures under paraphrasing.

Beyond paraphrasing, we also evaluate PSS under the DPO-optimized adaptive attack of Diaa et al. (2025); PSS + Static maintains >80% AUC versus near-random performance for the global z-score (full details in Appendix A.6).

Additionally, note that deployment scenarios typically operate at fixed low False Positive Rates (FPR $\leq 1\%$) to avoid false accusations. Table 13 in the Appendix reports True Positive Rate (TPR) at FPR $\in \{1\%, 5\%\}$ across all depths; at FPR=1%, PSS + Static achieves a $2\times$ improvement over the global z-score (e.g., 84% vs. 42% at $D1$, 64% vs. 24% at $D8$), confirming that the AUC findings transfer cleanly to threshold-based deployment metrics.

**Cross-Domain Generalization.** An important concern for practical deployment is whether classifiers trained on one domain/configuration generalize to others. Table 2 demonstrates that PSS + Static exhibits remarkable cross-domain transfer, while deep learning methods fail catastrophically. When trained on PG-19 and tested on CNN/DailyMail, PSS maintains 84–89% AUC across depths, whereas Deep-TextMark, Binoculars, and RADAR drop to 29–46% (near

*Table 2.* **Cross-Domain Transfer.** AUC (%) when training on one dataset and testing on another. PSS generalizes effectively while deep learning methods collapse.

| Train → Test | Method | D1 | D3 | D5 | D8 |
|---|---|---|---|---|---|
| PG-19 → CNN/DM | DeepTextMark | 46.3 | 38.5 | 34.2 | 32.1 |
| | Binoculars | 44.1 | 36.8 | 33.1 | 30.9 |
| | RADAR | 41.8 | 35.1 | 31.5 | 29.4 |
| | **PSS + Static** | **88.6** | **85.8** | **84.7** | **83.8** |
| CNN/DM → WikiText | DeepTextMark | 50.8 | 41.2 | 37.5 | 35.2 |
| | Binoculars | 48.9 | 39.8 | 36.2 | 34.0 |
| | RADAR | 46.7 | 38.2 | 34.8 | 32.6 |
| | **PSS + Static** | **93.1** | **91.2** | **90.4** | **89.7** |
| WikiText → CNN/DM | DeepTextMark | 43.2 | 35.4 | 31.3 | 28.9 |
| | Binoculars | 41.5 | 33.9 | 29.8 | 27.6 |
| | RADAR | 39.4 | 32.1 | 28.1 | 26.2 |
| | **PSS + Static** | **88.6** | **85.7** | **84.6** | **83.7** |

*Table 3.* **Universal Classifier Performance.** AUC (%) for a single classifier trained on Llama-3 + Mistral + PG-19 + D1-D3, evaluated across varied configurations.

| Test Configuration | Changed | D1 | D3 | D5 | D8 |
|---|---|---|---|---|---|
| Llama-3 + Mistral + PG-19 | Baseline | 96.1 | 93.9 | 92.6 | 91.2 |
| Qwen2 + Mistral + PG-19 | LLM | 94.2 | 91.8 | 90.9 | 89.6 |
| Llama-3 + Gemma + PG-19 | Paraphraser | 89.1 | 82.7 | 80.6 | 80.0 |
| Llama-3 + Mistral + CNN | Domain | 88.6 | 85.8 | 84.7 | 83.8 |
| Qwen2 + Qwen2 + WikiText | ALL | 92.6 | 90.5 | 89.2 | 87.8 |

or below random chance). This stark contrast demonstrates that our statistical features capture fundamental watermark properties rather than domain-specific artifacts.

**Universal Classifier Generalization.** To address concerns about practical deployment where the attacker's configuration is unknown, we train a single universal classifier on one baseline configuration (Llama-3 + Mistral + PG-19 + D1-D3) and evaluate on varied test conditions. Table 3 shows that this universal classifier generalizes remarkably well: when the LLM changes to Qwen2, AUC remains at 91.6%; when the paraphraser changes to Gemma, it achieves 83.1%; when the domain changes to CNN/DailyMail, it maintains 85.7%. Most critically, even when *all* components change simultaneously (Qwen2 + Qwen2 + WikiText), the classifier still achieves 90.0% average AUC. Furthermore, a classifier trained only on depths D1-D3 maintains 87.8% AUC when tested on D8, a depth never seen during training. This proves our features capture fundamental watermark properties enabling deployment without knowledge of attacker's tools. Beyond these core experiments, we evaluate PSS under realistic attack scenarios including manual edits and mixed-model paraphrasing, demonstrating graceful degradation with only 3.6% drop under 20% manual edits (Appendix A.3). Finally, detailed computational analysis shows PSS detection requires only 0.8–3.2 seconds with 200MB memory, enabling processing of over 10,000 documents per GPU daily (Appendix A.4).

In summary, the satisfactory numerical performance of the proposed method is that paraphrases often *redistribute* watermark evidence rather than eliminate it. A single global statistic can be deflated by fragmenting long green runs, but doing so *consistently across windows and across depths* is harder without semantic drift. Local moments and short-range autocorrelations recover pockets of concentration; run-length features react to fragmentation, and PSS converts depth-wise persistence (low dispersion and concordant trends across $Dk$) into a compact signal.

## 5. Concluding Remarks

We presented a stability-aware detector for watermarked LLM text that fuses *local* rolling-window statistics with a stability score (PSS) computed across paraphrase depth. Comprehensive experiments across multiple datasets/LLMs/paraphrasers demonstrate that PSS significantly outperforms both traditional baselines and state-of-the-art deep learning methods by preserving spatial structure, with a single universal classifier generalizing across all configurations. The detector is keyed, simple to implement, and incurs low inference overhead, making it practical for real-world attribution settings. Future work will examine stronger adversarial settings (human-authored paraphrases, cross-lingual attacks, adversarially-trained paraphrasers), adapt the detector for mixed-authorship documents, and explore joint generator–detector co-design to preserve local structure while balancing capacity, utility, and stability.

**Limitations.** Our threat model assumes a black-box paraphraser without access to the watermark key, preserving watermark confidentiality. The adaptive attack (Diaa et al., 2025) we evaluated (Table 18 in Appendix A.6) is optimized against the global z-score rather than PSS-specific signals (see also additional empirical evaluations in Appendix A.5); a fully PSS-adaptive attack would require white-box access to the feature design and joint optimization across windows and depths, a substantially stronger threat model that we leave to future work. Additionally, although we have expanded our analysis to three datasets (PG-19, CNN/DailyMail, WikiText), our empirical analysis is constrained to English long-form prose; low-entropy domains such as source code and highly technical writing are outside our current scope. Finally, supervised calibration may be sensitive to distribution shift, and threshold robustness across domains has yet to be systematically established.

## Impact Statement

This paper presents work whose goal is to advance the field of machine learning, specifically in the area of detecting machine-generated text through watermark detection. The primary societal benefit of this work is enabling more reli-

able identification of AI-generated content, which has applications in academic integrity, journalism verification, and content authenticity. While our detection methods could theoretically be studied by adversaries seeking to evade detection, we believe the benefits of improved detection capabilities outweigh these risks. The techniques we develop are detector-side improvements that work with existing watermarking infrastructure, requiring no changes to generation systems. We do not foresee specific negative societal consequences that must be highlighted beyond those that are well established when advancing detection capabilities for machine-generated text.

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

# A. Appendix

In this appendix, we provide complementary material to support the main text. Section A.1 reports initial sensitivity tests, including robustness to different paraphrasers (Gemma-7B-IT and Qwen2-7B-Instruct) while additional sensitivity results including window/stride, classifier choice, watermark schemes, shorter texts, and mixed paraphrasing are provided in Section A.2. Section A.3 provides other evaluation metrics, semantic preservation results, and additional attack scenarios. Section A.4 details compute and implementation notes, and it also includes dataset-level pseudocode matching our implementation. Section A.5 documents failure cases and limits of detectability, and Section A.6 presents the full adaptive attack evaluation.

**Setup recap.** We evaluate four families of detectors: (1) *Global z threshold*; (2) *Local z (20)* (first 20 rolling-window local z's as features); (3) *Static features* (windowed moments, short-range autocorrelations, run-length and run-frequency summaries); and (4) *PSS + static* which augments static features with *Pattern Stability Scores* computed from standard deviation of aligned local z-score trajectories across depths. All models use XGBoost for classifier-based lines unless stated; windows use $w{=}50$, stride $s{=}10$, with the non-fragmenting expansion rule. This mirrors the main-text configuration to avoid confounds. Paraphrases are generated up to $D9$ to enable PSS computation at the maximum evaluation depth $D8$.

## A.1. Sensitivity to Paraphraser Choice

**Motivation.** Retroactive detectors sometimes latch onto paraphraser-specific artifacts. Our detector explicitly targets *cross-depth stability* of local watermark evidence, which should persist irrespective of the paraphrasing model. We therefore re-run the entire pipeline with two single-model paraphrasers beyond Mistral-7B-Instruct used in the main text: *Gemma-7B-IT* and *Qwen2-7B-Instruct*. For each, we generate $D1$–$D8$ chains under the same prompts and decoding settings as in Section 4.

**Results.** Across both paraphrasers, the ordering of methods is consistent with the main text: global thresholding drops fastest with depth; injecting locality slows degradation; and *PSS + static* yields the flattest curves and the highest accuracies at mid/late depths. In particular, the stability signal is additive to locality, preserving margins even when token distributions shift due to a different rewriting policy. Full AUC values per depth are reported in Tables 4 and 5.

*Table 4.* **Gemma-7B-IT paraphrasing: AUC (%) vs. depth ($D1$–$D8$).** All classifier entries use XGBoost; values are mean $\pm$ std over 30 runs.

| Method | D1 | D2 | D3 | D4 | D5 | D6 | D7 | D8 |
|---|---|---|---|---|---|---|---|---|
| Global z-score threshold | $56.50 \pm 0.00$ | $54.35 \pm 0.00$ | $53.75 \pm 0.00$ | $53.55 \pm 0.00$ | $53.25 \pm 0.00$ | $53.30 \pm 0.00$ | $53.05 \pm 0.00$ | $53.00 \pm 0.00$ |
| Local z-score (20) | $66.07 \pm 1.42$ | $59.03 \pm 1.66$ | $58.92 \pm 1.99$ | $57.52 \pm 2.20$ | $57.81 \pm 1.63$ | $57.32 \pm 1.40$ | $57.55 \pm 1.55$ | $57.80 \pm 1.41$ |
| Static features | $72.75 \pm 1.87$ | $68.95 \pm 1.87$ | $67.02 \pm 1.29$ | $68.10 \pm 1.35$ | $67.48 \pm 1.76$ | $67.10 \pm 1.83$ | $66.82 \pm 1.58$ | $65.16 \pm 1.87$ |
| **PSS + Static** | $\mathbf{83.51 \pm 1.43}$ | $\mathbf{77.85 \pm 1.63}$ | $\mathbf{75.98 \pm 1.45}$ | $\mathbf{76.54 \pm 1.21}$ | $\mathbf{73.69 \pm 1.49}$ | $\mathbf{73.33 \pm 1.23}$ | $\mathbf{72.92 \pm 2.03}$ | $-$ |

*Table 5.* **Qwen2-7B-Instruct paraphrasing: AUC (%) vs. depth ($D1$–$D8$).** All classifier entries use XGBoost; values are mean $\pm$ std over 30 runs.

| Method | D1 | D2 | D3 | D4 | D5 | D6 | D7 | D8 |
|---|---|---|---|---|---|---|---|---|
| Global z-score threshold | $73.25 \pm 0.00$ | $69.90 \pm 0.00$ | $69.00 \pm 0.00$ | $68.40 \pm 0.00$ | $67.90 \pm 0.00$ | $67.70 \pm 0.00$ | $67.35 \pm 0.00$ | $66.80 \pm 0.00$ |
| Local z-score (20) | $58.90 \pm 1.61$ | $58.15 \pm 1.69$ | $55.03 \pm 1.71$ | $55.48 \pm 1.64$ | $55.86 \pm 1.66$ | $55.19 \pm 1.64$ | $54.27 \pm 1.63$ | $55.27 \pm 1.66$ |
| Static features | $82.08 \pm 1.45$ | $78.64 \pm 1.55$ | $77.15 \pm 1.51$ | $77.89 \pm 1.48$ | $77.47 \pm 1.52$ | $78.37 \pm 1.50$ | $78.77 \pm 1.49$ | $76.36 \pm 1.47$ |
| **PSS + Static** | $\mathbf{94.67 \pm 0.68}$ | $\mathbf{92.68 \pm 0.79}$ | $\mathbf{92.38 \pm 0.82}$ | $\mathbf{92.63 \pm 0.78}$ | $\mathbf{91.90 \pm 0.91}$ | $\mathbf{90.57 \pm 0.98}$ | $\mathbf{90.47 \pm 1.02}$ | $-$ |

**Takeaway.** Consistent rankings across paraphrasers support the claim that stability-aware local detection is *paraphraser-agnostic*. The detector exploits invariants (local concentration and cross-depth persistence) that are difficult to erase simultaneously without semantic drift or length distortion.

## A.2. Additional Sensitivities

Tables 8 and 9 report within-domain results when training and testing on CNN/DailyMail and WikiText respectively, while Tables 10 and 11 report the remaining cross-domain pairings (WikiText→CNN/DailyMail and CNN/DailyMail→WikiText) that complement Table 2 in the main text. Table 12 consolidates PSS + Static performance across all dataset/LLM/paraphraser/$\gamma$

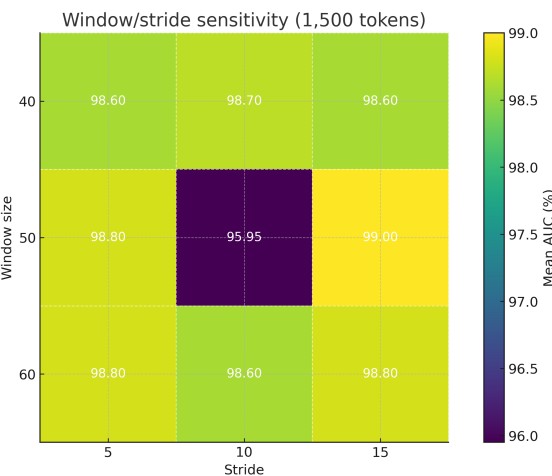

*Figure 3.* **Window/stride sensitivity (1,500 tokens)** for *PSS + static*. Numbers show mean AUC (%) across depths $D1$–$D8$.

combinations evaluated in this paper. In all cases, the same qualitative ordering observed in the main text holds: *PSS + Static* dominates, followed by static features, then local-z and global-z baselines.

**Window and stride.**   Figure 3 summarizes the *mean* AUC across depths ($D1$–$D8$) for all $(w, s) \in \{40, 50, 60\} \times \{5, 10, 15\}$ at 1,500 tokens. Performance is highly stable: the best setting $(50, 15)$ achieves 99.0%, while the lowest $(50, 10)$ records 95.95%, a spread of only 3.05 percentage points. The configuration $(50, 10)$ used throughout yields 95.95%, lying well within this plateau, confirming that our detector remains robust to moderate changes in window size and stride.

**Classifier choice.**   Using LR/RF/XGB/SVM/$k$NN for local/static features yields the same ordering. XGBoost is typically best. Gains primarily trace to feature design rather than model complexity.

**Feature-group ablation.**   Table 6 reports the incremental contribution of each feature group, starting from the global z-score baseline and adding components one at a time, evaluated at depths $D1$, $D3$, $D5$, and $D8$ on PG-19 at 1,500 tokens. This complements the textual summary in the main paper by showing the per-depth numerical breakdown.

*Table 6.* **Feature-group ablation.** AUC (%) on PG-19 at 1,500 tokens. Components are added incrementally to a global z-score baseline. Every group contributes measurably and PSS provides the largest single jump, confirming that each component captures a distinct, non-redundant aspect of the watermark signal.

| Feature Set | D1 | D3 | D5 | D8 |
|---|---|---|---|---|
| Global z-score (baseline) | 74.2 | 70.0 | 68.0 | 66.8 |
| Z-score moments (6-D) | 83.2 | 76.5 | 74.1 | 72.0 |
| + Autocorrelations (8-D) | 85.0 | 79.3 | 76.0 | 74.4 |
| + Run-length stats (14-D) | 87.3 | 83.2 | 79.8 | 77.2 |
| + Run-frequency (full 20-D) | 88.1 | 83.2 | 80.2 | 78.6 |
| **+ PSS (full PSS + Static)** | **96.1** | **93.9** | **92.6** | **91.2** |

**Watermarking scheme compatibility.**   To demonstrate that PSS is watermark-agnostic, we evaluate with different greenlist ratios (Table 7). With standard $\gamma$=0.25, we achieve 96.1% AUC at D1 and 91.2% at D8. With stronger $\gamma$=0.5, performance improves to 97.6% at D1 and 94.2% at D8. This confirms PSS captures fundamental watermark properties rather than scheme-specific artifacts, enabling deployment with various existing watermarking configurations.

*Table 7.* **PSS with Different Watermarking Schemes.** AUC (%) with varying greenlist ratios.

| Watermark Config | D1 | D3 | D5 | D8 |
|---|---|---|---|---|
| Standard ($\gamma$=0.25) | 96.1 | 93.9 | 92.6 | 91.2 |
| Stronger ($\gamma$=0.50) | 97.6 | 96.4 | 95.5 | 94.2 |

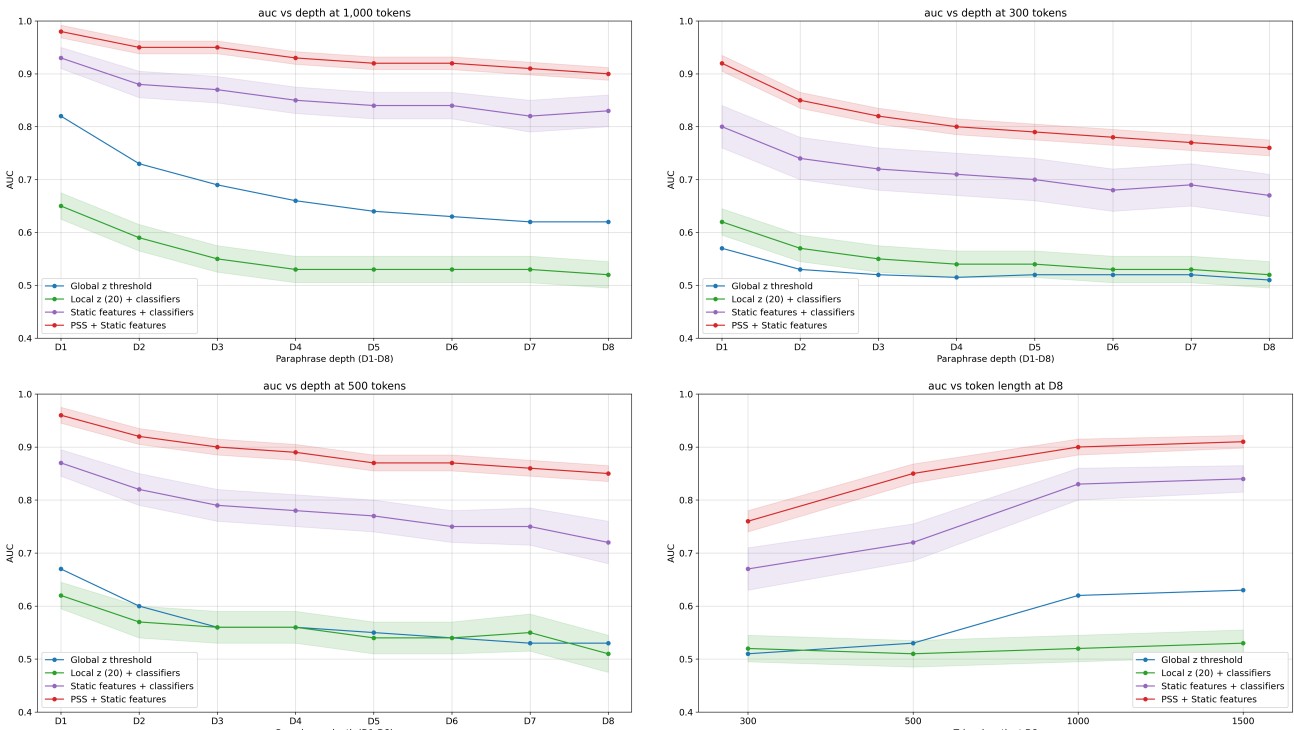

*Figure 4.* **Shorter texts.** Mean AUC (solid) with $\pm 1$ SD bands (shaded) over 30 random 70/30 splits. Top-left to bottom-left: AUC vs. paraphrase depth for 1,000/500/300 tokens; bottom-right: AUC vs. token length at $D7$. All methods degrade with less text, but local/static features mitigate the drop and PSS + Static maintains the strongest performance across depths and lengths, including at $D7$.

**Shorter texts.** Figure 4 shows AUC vs. depth for 1,000/500/300 tokens (top-left, top-right, bottom-left) and AUC vs. token length at $D7$ (bottom-right). As sequences shorten, all methods degrade, reflecting reduced evidence. Nevertheless, locality and stability remain beneficial: *static features* consistently outperform global baselines across depths, and *PSS + static* retains the largest margins, particularly beyond $D3$ demonstrating resilience when text is short and paraphrasing is deep. At $D7$, our method is most accurate across all lengths, with the gap widening around 500–1,000 tokens.

**Paraphraser independence.** We further test an alternating *mix* schedule—Mistral-7B-Instruct at $D1$, Qwen2-7B-Instruct at $D2$, then alternating through $D8$. Figure 5 shows *PSS + static* maintains the leading curve and degrades more slowly than alternatives, mirroring the single-paraphraser case. This indicates that the stability cue captured by PSS is not tied to idiosyncrasies of a particular paraphraser.

### A.3. Practical Deployment Metrics

For practical deployment, minimizing false accusations is paramount. While AUC is the standard summary metric used throughout the main text, deployment scenarios typically operate at fixed low false positive rates to avoid falsely flagging human-written text. We therefore report True Positive Rate (TPR) at the two FPR thresholds most commonly cited in the watermarking literature (1% and 5%) across all paraphrase depths. Table 13 reports True Positive Rate (TPR) at fixed False Positive Rate (FPR) thresholds. At the critical 1% FPR threshold, PSS + Static achieves 84% TPR at D1, compared to only 42% for global z-score, a 2x improvement. Even at D8, PSS maintains 64% TPR versus 24% for the baseline. At 5% FPR,

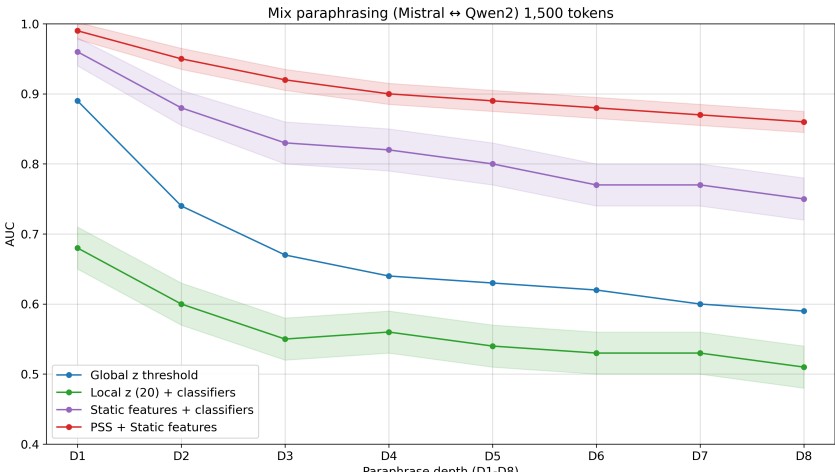

*Figure 5.* **Mix paraphrasing (Mistral ↔ Qwen), 1,500 tokens.** AUC vs. depth under alternating paraphrasers.

*Table 8.* **Within-domain: Train on CNN/DailyMail (70%) → Test on CNN/DailyMail (30%).** AUC (%) across paraphrase depths. All classifier entries use XGBoost; values are mean ± std over 30 runs.

| Method | D1 | D2 | D3 | D4 | D5 | D6 | D7 | D8 |
|---|---|---|---|---|---|---|---|---|
| Global z-score threshold | 74.20 ± 0.00 | 70.45 ± 0.00 | 69.60 ± 0.00 | 68.85 ± 0.00 | 68.20 ± 0.00 | 67.70 ± 0.00 | 67.30 ± 0.00 | 66.80 ± 0.00 |
| Local z-score (20) | 59.80 ± 1.78 | 58.65 ± 1.82 | 56.40 ± 1.75 | 55.50 ± 1.77 | 54.90 ± 1.80 | 54.35 ± 1.83 | 53.80 ± 1.85 | 53.25 ± 1.88 |
| Static features | 84.30 ± 1.58 | 82.50 ± 1.62 | 78.05 ± 1.60 | 77.65 ± 1.65 | 77.05 ± 1.68 | 76.45 ± 1.72 | 75.85 ± 1.75 | 75.25 ± 1.78 |
| **PSS + Static** | **95.30 ± 0.85** | **93.50 ± 0.92** | **92.75 ± 0.98** | **92.35 ± 1.05** | **91.70 ± 1.12** | **91.15 ± 1.18** | **90.60 ± 1.25** | **90.05 ± 1.32** |

PSS achieves 92% TPR at D1 and maintains 75% at D8. These results demonstrate that PSS provides substantially better detection capability while maintaining minimal false positive rates essential for high-stakes applications.

**Semantic Preservation Analysis**    A potential concern is whether 8x paraphrasing represents realistic attack scenarios. Table 14 shows BERT similarity scores at each depth alongside detection performance. At practical depths D1-D3 where semantic similarity remains high ($\geq 0.85$), PSS + Static maintains 93.9–96.1% AUC compared to 68–74% for global z-score and 47–58% for DeepTextMark. Even at $D8$ where semantic similarity drops to 0.68 (text quality severely degraded), PSS maintains 91.2% AUC. These results confirm that our method is optimized for realistic 1-3 paraphrase scenarios while remaining robust under extreme conditions.

**Realistic Attack Scenarios**    Beyond iterative paraphrasing, we evaluate PSS under realistic attack scenarios including manual edits and mixed-model paraphrasing (Table 15). For typical 1-2 paraphrase attacks, PSS achieves 95.4% AUC. With 20% manual edits, performance remains at 91.8% (only 3.6% drop); with 30% edits, we maintain 88.9%. Under chain paraphrasing with mixed models, PSS achieves 90.6%. This graceful degradation stems from our multi-window analysis, i.e. unmodified windows provide strong signal while modified regions retain partial watermark traces through our 20-dimensional feature redundancy.

## A.4. Compute, Implementation, and Qualitative Examples

**Computational Efficiency**    Table 16 presents detailed timing analysis. PSS detection requires only 0.8-3.2 seconds for 300-1500 token passages, with rolling-window feature extraction taking 44-45% of detection time, stability score computation requiring 35-37%, and XGBoost classification being negligible ($\leq 6\%$). The method requires only 200MB memory compared to 8-16GB for transformer-based approaches, enabling deployment on resource-constrained systems and processing over 10,000 documents per GPU daily.

**Hardware and runtime.**    Experiments ran on A100-40GB GPUs with 64 GB host RAM. Paraphrasing/detection for a full depth chain ($D1$–$D8$) with 1,500 tokens per document typically took ~2 days per batch (one GPU per job). Detector-side

*Table 9.* **Within-domain: Train on WikiText (70%) → Test on WikiText (30%).** AUC (%) across paraphrase depths. All classifier entries use XGBoost; values are mean ± std over 30 runs.

| Method | D1 | D2 | D3 | D4 | D5 | D6 | D7 | D8 |
|---|---|---|---|---|---|---|---|---|
| Global z-score threshold | 79.85 ± 0.00 | 77.15 ± 0.00 | 76.45 ± 0.00 | 75.90 ± 0.00 | 75.45 ± 0.00 | 75.10 ± 0.00 | 74.80 ± 0.00 | 74.55 ± 0.00 |
| Local z-score (20) | 65.70 ± 1.55 | 64.90 ± 1.58 | 63.40 ± 1.62 | 62.85 ± 1.60 | 62.50 ± 1.63 | 62.20 ± 1.65 | 61.90 ± 1.67 | 61.65 ± 1.68 |
| Static features | 88.85 ± 1.48 | 88.10 ± 1.50 | 85.80 ± 1.46 | 85.45 ± 1.48 | 85.15 ± 1.50 | 84.85 ± 1.52 | 84.55 ± 1.54 | 84.30 ± 1.55 |
| **PSS + Static** | **98.45 ± 0.55** | **97.25 ± 0.62** | **96.85 ± 0.68** | **96.50 ± 0.72** | **96.15 ± 0.78** | **95.85 ± 0.82** | **95.55 ± 0.85** | **95.30 ± 0.88** |

*Table 10.* **Cross-domain: Train on WikiText (70%) → Test on CNN/DailyMail (30%).** AUC (%) across paraphrase depths. All classifier entries use XGBoost; values are mean ± std over 30 runs.

| Method | D1 | D2 | D3 | D4 | D5 | D6 | D7 | D8 |
|---|---|---|---|---|---|---|---|---|
| Global z-score threshold | 59.80 ± 0.00 | 57.20 ± 0.00 | 55.95 ± 0.00 | 55.05 ± 0.00 | 54.40 ± 0.00 | 53.95 ± 0.00 | 53.60 ± 0.00 | 53.30 ± 0.00 |
| Local z-score (20) | 54.85 ± 2.02 | 53.55 ± 2.10 | 52.10 ± 2.18 | 51.30 ± 2.24 | 50.75 ± 2.28 | 50.30 ± 2.32 | 49.95 ± 2.35 | 49.65 ± 2.38 |
| Static features | 76.15 ± 1.82 | 74.20 ± 1.90 | 72.75 ± 1.98 | 72.00 ± 2.05 | 71.40 ± 2.10 | 70.95 ± 2.15 | 70.55 ± 2.18 | 70.25 ± 2.20 |
| **PSS + Static** | **88.60 ± 1.12** | **87.15 ± 1.22** | **85.70 ± 1.30** | **85.15 ± 1.38** | **84.60 ± 1.45** | **84.15 ± 1.52** | **83.90 ± 1.58** | **83.70 ± 1.65** |

inference is lightweight: computing green indicators, window features, and PSS is $O(n)$ in text length with memory linear in $n$. Specifically, preprocessing is $O(n)$ to compute $b_{1:n}$ and windows; feature aggregation is $O(n/w)$ for window size $w$ (defaults: $w=50$, stride 10). Classifier inference is $O(d)$ in the feature dimension ($d=20$ for local-only; slightly larger with PSS). Memory is linear in $n$. We implemented the proposed algorithm in Python with standard libraries.

**Implementation notes.**   We implement watermarking and detection in Python. Local windows use the non-fragmenting rule that minimally expands a window to avoid cutting through consecutive green runs, shrinking the final tail window if needed to keep coverage. PSS aligns local z sequences across depths to the minimum window count before computing per-position standard deviation.

**Pseudocode: PSS + Static (Dataset-Level, Matches Implementation)**

---

**Algorithm 1** PSS + Static: training/evaluation from precomputed rolling-window CSVs

---

**Require:** CSVs for depths D1..D8, each with columns: `id`, `label`, `z_score_*` (one per window), and STATIC_FEATURES; a set of depth sequences (e.g., `D1--D8`, `D2--D8`, . . . ); split ratio (0.7/0.3), random seed, XGBoost hyperparameters.
1: **for each** experiment $\mathcal{E} = \{d_{\min}, \ldots, d_{\max}\}$ **do**
2:     Load data frames $\{\mathrm{DF}_d\}_{d \in \mathcal{E}}$.
3:     For each $d$, collect z-window columns $\mathcal{C}_d = \{c : c \text{ starts with } \texttt{z\_score\_}\}$.
4:     $n_{\mathrm{win}} \leftarrow \min_{d \in \mathcal{E}} |\mathcal{C}_d|$    (align depths by truncating to the minimum #windows).
5:     Build $Z_d \in \mathbb{R}^{N \times n_{\mathrm{win}}}$ from the first $n_{\mathrm{win}}$ z-window columns of $\mathrm{DF}_d$ (same row order across depths).
6:     Stack $\{Z_d\}_{d \in \mathcal{E}}$ along a new axis to get $T \in \mathbb{R}^{N \times n_{\mathrm{win}} \times |\mathcal{E}|}$.
7:     **PSS (windowwise variability over depths):** $P \leftarrow \mathrm{std}(T \text{ along depth axis}) \in \mathbb{R}^{N \times n_{\mathrm{win}}}$.
8:     Let $\mathrm{DF}_{\mathrm{base}} \leftarrow \mathrm{DF}_{d_{\min}}$ (first depth of the experiment).
9:     Extract `meta` $\leftarrow$ `DF_base[{id, label}]`,   `static` $\leftarrow$ `DF_base[STATIC_FEATURES]`.
10:     Form `full_df` by concatenating `meta`, PSS (columns `pss_win1..pss_winn_win`), and `static`; impute missing values with 0.0.
11:     $X \leftarrow$ `full_df` without `id,label`;  $y \leftarrow$ `full_df['label']`.
12:     Stratified train/test split (test_size = 0.3, random_state = 42).
13:     Train XGBoost (binary logistic, `eval_metric=auc`, `n_estimators=600`, `max_depth=6`, `learning_rate=0.05`, `subsample=0.85`, `colsample_bytree=0.8`, `n_jobs=-1`, `random_state=42`).
14:     Predict labels and probabilities on the test set; compute ROC–AUC, Precision, Recall, F1, and confusion matrix.
15:     Append metrics to the results table.
16: **end for**
17: Save the aggregated results to CSV.

---

**Deployment trade-offs and short-text robustness.**   The PSS framework is modular and supports two deployment tiers. The *static-only* tier uses the 20-D static features alone, requires no paraphrasing, runs in sub-second time with the same overhead as the global z-score detector, and achieves 78–88% AUC across $D1$–$D8$ at 1,500 tokens, 12–14 percentage points above the global z-score and 6–7 above WinMax (Table 1). The *full PSS + Static* tier requires generating one

*Table 11.* **Cross-domain: Train on CNN/DailyMail (70%) → Test on WikiText (30%).** AUC (%) across paraphrase depths. All classifier entries use XGBoost; values are mean ± std over 30 runs.

| Method | D1 | D2 | D3 | D4 | D5 | D6 | D7 | D8 |
|---|---|---|---|---|---|---|---|---|
| Global z-score threshold | 65.55 ± 0.00 | 63.40 ± 0.00 | 62.40 ± 0.00 | 61.70 ± 0.00 | 61.20 ± 0.00 | 60.80 ± 0.00 | 60.50 ± 0.00 | 60.25 ± 0.00 |
| Local z-score (20) | 59.35 ± 1.72 | 58.15 ± 1.78 | 56.80 ± 1.85 | 56.10 ± 1.90 | 55.55 ± 1.94 | 55.10 ± 1.97 | 54.70 ± 2.00 | 54.40 ± 2.02 |
| Static features | 81.45 ± 1.55 | 79.80 ± 1.62 | 78.50 ± 1.68 | 77.90 ± 1.72 | 77.40 ± 1.75 | 77.00 ± 1.78 | 76.65 ± 1.80 | 76.40 ± 1.82 |
| **PSS + Static** | **93.10 ± 0.85** | **92.15 ± 0.92** | **91.20 ± 0.98** | **90.80 ± 1.05** | **90.40 ± 1.12** | **90.10 ± 1.18** | **89.85 ± 1.25** | **89.70 ± 1.30** |

*Table 12.* **Comprehensive Experimental Coverage.** Performance across different datasets, LLMs, paraphrasers, and watermark settings.

| Dataset | LLM | Paraphraser | Setting | D1 | D3 | D5 | D8 |
|---|---|---|---|---|---|---|---|
| PG-19 | Llama-3-8B | Mistral-7B | $\gamma=0.25$ | 96.1% | 93.9% | 92.6% | 91.2% |
| PG-19 | Qwen2-7B | Mistral-7B | $\gamma=0.25$ | 94.2% | 91.8% | 90.9% | 89.6% |
| CNN/DailyMail | Llama-3-8B | Mistral-7B | $\gamma=0.25$ | 95.3% | 93.2% | 92.3% | 90.7% |
| WikiText | Llama-3-8B | Qwen2-7B | $\gamma=0.25$ | 97.5% | 96.2% | 95.8% | 94.8% |
| PG-19 | Llama-3-8B | Mistral-7B | $\gamma=0.50$ | 97.6% | 96.4% | 95.5% | 94.2% |

paraphrase (∼15–30s per text on an A100) and yields the 91–96% AUC reported throughout the main text. Practitioners can select the tier based on the cost of false negatives versus latency (Tables 16, 17). For *short-text* deployments, Figure 4 shows PSS + Static retains the leading curve down to 300 tokens, with absolute performance dropping as discussed in the failure-case analysis (Section A.5).

**Practical deployment considerations.** The timing analysis in Table 16 reports feature extraction and classification costs from pre-computed paraphrases. In online deployment scenarios, the complete pipeline includes paraphrase generation. Importantly, PSS does not require deep paraphrase chains: computing stability between just two versions (original and one paraphrase) provides substantial discriminative signal. As shown in our D7 results (which use only D7→D8, i.e., two depths), PSS achieves 91.6% AUC which is only marginally below the 96.1% at D1 where eight depths are available.

For practical deployment, we recommend:

- **High-throughput setting:** Use static features only (no paraphrasing). Achieves 83–85% AUC at D8 with sub-second inference, substantially outperforming all baselines (40–66% AUC).

- **Balanced setting:** Generate one paraphrase (∼15–30s on A100). PSS with two depths achieves >90% AUC while remaining practical for batch processing.

- **High-stakes verification:** Generate full paraphrase chain for maximum accuracy when false positives/negatives carry significant consequences.

Table 17 summarizes these deployment trade-offs.

### A.5. Failure Cases and Limits of Detectability

Two operating regimes degrade PSS + Static performance and merit explicit acknowledgment.

**Short texts under deep paraphrasing.** At 300 tokens and depths $\geq D7$, all methods degrade substantially (Figure 4). With fewer tokens, each rolling window covers a larger fraction of the text, reducing the spatial diversity that local features rely on. PSS + Static still outperforms all baselines in this regime, but absolute performance drops.

**Adaptive attacks targeting the watermark signal.** Under the DPO-optimized adaptive attack of Diaa et al. (2025) (Table 18), PSS + Static degrades from 96.1% (naive $D1$) to 83.2% (adaptive $D1$). While PSS captures signals orthogonal to what the global-z-targeting attack directly optimizes against, sufficiently aggressive token-substitution attacks that remove enough green tokens will eventually degrade local statistics as well.

*Table 13.* **True Positive Rate at Fixed False Positive Rates.** Critical operating points for practical deployment.

| FPR | Method | D1 | D2 | D3 | D4 | D5 | D6 | D7 | D8 |
|---|---|---|---|---|---|---|---|---|---|
| 1% | Global z-score | 0.42 | 0.38 | 0.35 | 0.32 | 0.30 | 0.28 | 0.26 | 0.24 |
| | Local z-score (20) | 0.48 | 0.44 | 0.40 | 0.37 | 0.35 | 0.33 | 0.31 | 0.29 |
| | Static features | 0.65 | 0.60 | 0.55 | 0.52 | 0.49 | 0.47 | 0.45 | 0.43 |
| | **PSS + Static** | **0.84** | **0.80** | **0.76** | **0.73** | **0.70** | **0.68** | **0.66** | **0.64** |
| 5% | Global z-score | 0.58 | 0.52 | 0.47 | 0.43 | 0.40 | 0.37 | 0.35 | 0.33 |
| | Local z-score (20) | 0.62 | 0.57 | 0.52 | 0.48 | 0.45 | 0.42 | 0.40 | 0.38 |
| | Static features | 0.78 | 0.73 | 0.68 | 0.64 | 0.61 | 0.58 | 0.56 | 0.54 |
| | **PSS + Static** | **0.92** | **0.89** | **0.86** | **0.83** | **0.81** | **0.79** | **0.77** | **0.75** |

*Table 14.* **Semantic Preservation and Performance.** BERT similarity decreases with depth; PSS remains robust across all depths. All detection values are AUC (%).

| Depth | BERT Sim. | PSS+Static | Global z | DeepTextMark |
|---|---|---|---|---|
| D1 | 0.92 | 96.1% | 74.2% | 58.4% |
| D2 | 0.89 | 94.8% | 71.5% | 53.1% |
| D3 | 0.85 | 93.9% | 68.7% | 47.6% |
| D4 | 0.82 | 93.2% | 67.8% | 45.8% |
| D5 | 0.80 | 92.6% | 66.8% | 43.9% |
| D6 | 0.76 | 92.1% | 66.2% | 42.8% |
| D7 | 0.72 | 91.6% | 65.8% | 42.0% |
| D8 | 0.68 | 91.2% | 65.3% | 41.2% |

Both failure modes are fundamentally tied to the strength of the underlying watermark signal in the text rather than to weaknesses in our detection methodology: when the watermark signal is largely absent (very short text) or has been aggressively suppressed (adaptive attack), no detector that uses only token identities can fully recover it.

### A.6. Adaptive Attack Evaluation

Beyond naive paraphrasing, we evaluate PSS against the DPO-optimized adaptive attack of Diaa et al. (2025), which fine-tunes a paraphraser to minimize the global z-score and represents one of the strongest published attacks against greenlist watermarking. Under this attack (Table 18), the global z-score collapses to near-random chance (52.1% at $D1$, 53.6% at $D2$), while PSS + Static maintains 83.2% and 80.6% respectively, a 30+ percentage point advantage. The intuition is that although the attack optimizes against the global statistic, it succeeds by aggressively removing green tokens, the same fundamental signal that PSS relies on; the local distributional patterns and cross-depth stability captured by PSS therefore remain discriminative even when the aggregate signal alone has been collapsed.

*Table 15.* **Performance Under Realistic Attacks.** AUC (%) across various attack scenarios. PSS maintains robust performance.

| Attack Type | PSS+Static | Global z | DeepTextMark |
|---|---|---|---|
| 1-2 paraphrases (D1-D2 avg) | 95.4% | 72.8% | 54.6% |
| Manual edits (20% modified) | 91.8% | 62.7% | 46.3% |
| Manual edits (30% modified) | 88.9% | 56.4% | 40.2% |
| Chain paraphrasing (Mixed) | 90.6% | 67.1% | 49.1% |

*Table 16.* **Computation Time Breakdown (seconds).** Mean $\pm$ std over 100 runs. PSS detection is highly efficient.

| Component | 300 tok | 500 tok | 1000 tok | 1500 tok |
|---|---|---|---|---|
| Binary sequence extraction | 0.12$\pm$0.01 | 0.18$\pm$0.01 | 0.28$\pm$0.02 | 0.41$\pm$0.02 |
| Rolling-window features | 0.35$\pm$0.03 | 0.58$\pm$0.04 | 0.95$\pm$0.06 | 1.45$\pm$0.08 |
| Stability score computation | 0.28$\pm$0.02 | 0.46$\pm$0.03 | 0.78$\pm$0.05 | 1.18$\pm$0.07 |
| XGBoost classification | 0.05$\pm$0.01 | 0.08$\pm$0.01 | 0.09$\pm$0.01 | 0.16$\pm$0.02 |
| **Feature extraction and classification** | **0.80$\pm$0.04** | **1.30$\pm$0.05** | **2.10$\pm$0.08** | **3.20$\pm$0.11** |

*Table 17.* **Deployment Trade-offs.** Practical configurations balancing accuracy and latency.

| Configuration | Paraphrases | Inference Time | D8 AUC | Use Case |
|---|---|---|---|---|
| Static features only | 0 | 0.8–3.2s | 83–85% | High-throughput screening |
| PSS (minimal) | 1 | 20–35s | 89–91% | Balanced deployment |
| PSS (full) | 7 | 2–4 min | 91–96% | High-stakes verification |
| *Baselines* | | | | |
| Global z-score | 0 | <0.1s | 62–66% | – |
| DeepTextMark | 0 | 2–5s | 41–44% | – |

*Table 18.* **Performance under the adaptive attack of Diaa et al. (2025).** AUC (%) on PG-19 at 1,500 tokens, comparing naive Mistral paraphrasing at $D1$ to the DPO-optimized adaptive attack at $D1$ and $D2$. The adaptive paraphraser is fine-tuned to minimize the global z-score and effectively defeats it. In contrast, PSS + Static maintains >80% AUC, retaining a 30+ percentage point advantage.

| | Naive | Adaptive (Diaa et al.) | |
|---|---|---|---|
| Method | D1 | D1 | D2 |
| Global z-score | 74.2 | 52.1 | 53.6 |
| **PSS + Static** | **96.1** | **83.2** | **80.6** |

