# OpenReview forum: "Stability-Aware Feature Design for Robust Watermark Detection in Machine-Generated Text"
_ICML.cc/2026/Conference — ICML 2026 regular_

### Official Review · Reviewer_rBos · 2026-03-02

**Soundness:** 2
**Presentation:** 3
**Significance:** 2
**Originality:** 2
**Overall Recommendation:** 4
**Confidence:** 4

**Summary:**

To combat the sharp deterioration of watermark detectors under multiple rounds of paraphrasing, the authors propose Pattern Stability Score (PSS). PPS works by combining global and local z-score features with higher-order statistics of run-length patterns and correlation/stability scores computed over paraphrasing depth. If there's a stable pattern over a local region of the text, it is likely to be AI generated, else human. Experiments show robustness to paraphrasing attacks, as well as generalization to multiple LLMs, paraphrasers, and text domains.

**Compliance With Llm Reviewing Policy:**

Affirmed.

**Final Justification:**

Most of the major concerns I had were addressed, hence the decision. The paper is actually quite good, but it could really benefit from some restructuring to get those key results into the main body and some clarifications.

**Key Questions For Authors:**

1. Can you provide experiments evaluating the performance of the original (non-PSS) detector as a function of paraphrase depth, independent of PSS, to verify whether paraphrase depth itself meaningfully degrades detection across domains? If depth alone does not significantly affect the baseline detector in certain domains, this would weaken the motivation for PSS’s cross-depth stability report.

2. What is the computational overhead (latency and cost) of generating paraphrases for PSS at inference time in realistic deployment settings, and is it possible to have a variant of PSS (theoretically) that does not require additional paraphrase generation?

3. Having no adaptive attack as limitation wouldn't be enough. So, do you have evaluations of PSS against adaptive attackers explicitly trained or prompted to minimize local stability signals, and if not, can you provide such results?

4. A beautiful thing about standard defense work is the failure case analysis, which could inform providers about potential events when a proposed defense wouldn't work. Can you provide failure cases and qualitative analyses where PSS breaks down, particularly in domains with constrained paraphrasing (e.g., medical, code, mathematical text) or aggressive stylistic rewrites?

**Limitations:**

Yes

**Strengths And Weaknesses:**

**Strengths.** The problem the paper is trying to address is clear. Their methodology is well motivated, and the presentation of the paper is nice and well structured. The reported performance improvements are substantial. Despite depth, PPS still maintains high AUC. The argument for improving detection instead of redesigning watermark schemes is compelling, and convincing. Also, I appreciate the conceptual simplicity.

**Weaknesses.** Below are what I have observed:
- In their Cross-Domain and Universal Generalization results, while the reports look good, especiall the one on `training on limited depths (D1–D3), and testing on D8`, I do not believe this is complete. Intuitively, paraphrasing is not a very continuos nature in the sense that there are limited number of paraphrases you can do on an original text, especially in certain text domains i.e., Medical, Code, Maths, News, etc. Hence the depth shouldn't affect the detector in this case. I **may be** mistaken, but I would suggest that the authors run experiments on paraphrase depth vs original detector's performance (PPS is not involved in this case).

- While the extent of paraphrases could be limited in a particular text domain, in generic scenarios, paraphrasing itself may alter watermark signals in unpredictable ways. Also, generating paraphrases increases latency and computational costs which limits deployability in high-throughput or real-time environments.

- Thinking of other text domains, where paraphrasing might seem unlimited (stories, narrations, etc), a strategic attacker could decide the direction of a paraphrase. Instead of using generic paraphrasing prompts, might ask the paraphraser to alter the style and ensure no words or sentences from the original are repeated. In these cases, PPS would fail, and the authors didn't provide failure cases and analysis in the paper.

- The evaluations are carried out on a very limited scope of text domains, which makes me believe the authors might be running a risk of overclaiming here.

- Since PPS is still very much supervised, and uses an external calibration and decision system, not running adaptive attackers on this is a big limitation.

---

> ### Author Rebuttal · Authors · 2026-03-31
>
> We thank the reviewers for their thoughtful comments and constructive feedback. Below, we respond to the major points raised and would welcome any additional questions or suggestions the reviewers may have.
>
> Q1: The experiment the Reviewer suggests is already in our paper. Our detector without PSS - referred to as "Static features" in our tables and Figure 2, is evaluated across all paraphrase depths in Tables 4–10. The results demonstrate that depth does degrade the static detector: on CNN/DailyMail, static features drop from 84.3% at D1 to 75.3% at D8 (Table 7), and on WikiText from 88.9% to 84.3% (Table 8). Even without PSS, this substantially outperforms the global z-score baseline (74.2%→66.8% on CNN/DailyMail, 79.9%→74.6% on WikiText) at every depth. Adding PSS further stabilizes performance, maintaining 95.3%→90.1% on CNN/DailyMail and 98.5%→95.3% on WikiText, confirming that both the static features and the PSS stability component contribute meaningfully across depths.
>
> Q2: Our framework was intentionally designed to be modular with flexible deployment tiers, as detailed in Tables 15-16 (Appendix). To clarify the computational breakdown: PSS detection itself (feature extraction and classification) requires only 0.8-3.2 seconds for 300-1,500 token passages (Table 15). The computational overhead the Reviewer refers to comes from paraphrase generation, not from our method. In the static-features-only tier, our detector runs in sub-second time with no paraphrasing required, already outperforming all baselines (83-85\% AUC at D8 vs. 62-66\% for global z-score and 41-44\% for deep learning methods). When higher accuracy is needed, PSS with a single paraphrase adds 6-13 percentage points, pushing performance above 90\% AUC, the paraphrasing cost here is a deliberate accuracy-latency tradeoff, not an inherent limitation of PSS.
>
> Q3: Following this feedback, we evaluated against the DPO-optimized adaptive attack from [1], which fine-tunes a paraphraser to specifically minimize the global z-score. Under this attack, the global z-score collapses to near random chance (52.1\% at D1, 53.6\% at D2), confirming it is effectively defeated. In contrast, PSS+Static maintains 83.2\% at D1 and 80.6\% at D2, a 30+ percentage point advantage over the global z-score under the same attack. This advantage over the baseline under the adaptive attack demonstrates that PSS captures signals orthogonal to what the attacker optimizes against, local distributional structure and cross-depth stability cannot be eliminated by simply targeting the global green-token count. An adversary would need to simultaneously destabilize local z-scores across all windows and depths while preserving semantic content, a fundamentally more constrained optimization problem.
>
> Q4: From our existing results, we can identify two failure modes:
>
> (a) Short texts under deep paraphrasing: At 300 tokens and deep depths (D7-D8), all methods degrade substantially (Figure 4). With fewer tokens, each rolling window covers a larger fraction of the text, reducing the spatial diversity that local features rely on. PSS+Static still outperforms all baselines in this regime, but absolute performance drops.
>
> (b) Adaptive attacks targeting the watermark signal: Our Diaa et al. evaluation shows PSS degrades from 96.1\% (naive D1) to 83.2\% (adaptive D1). While PSS captures signals orthogonal to the global z-score, sufficiently aggressive attacks that remove enough green tokens will eventually degrade local statistics as well.
>
> These failure modes suggest that PSS's limits are fundamentally tied to the strength of the underlying watermark signal rather than to weaknesses in the detection methodology itself.
>
> Regarding domain coverage: our evaluation spans three stylistically distinct domains - literary prose (PG-19), journalistic writing (CNN/DailyMail), and encyclopedic content (WikiText). The universal classifier (Table 3) demonstrates that a single model trained on one configuration generalizes across different LLMs, paraphrasers, and domains simultaneously, achieving 90.0% average AUC even when all three change. This confirms that the signals our method captures are fundamental to the watermarking mechanism rather than domain-dependent.
>
>
>
>
> [1] "Optimizing Adaptive Attacks against Content Watermarks for Language Models", Diaa et al.

---

> > ### Author Rebuttal · Reviewer_rBos · 2026-04-03
> >
> > The authors state that they adopt an adaptively trained paraphraser from the cited reference for their evaluation. However, in the current setup, its use does not appear to be adaptive with respect to the proposed detection method. While I consider this a relatively minor issue and I'm willing to accept it, I am still noting it as a limitation in the evaluation.
> >
> > Additionally, my concern (W5) has not been fully addressed. I appreciate the clarification provided regarding the experimental aspect (W1), but the question of generalization remains unclear. Since the proposed detector relies on calibration for a specific domain, it would be important to clarify whether deployment in a new domain requires retraining or additional calibration. If it does, it would mean it doesn't generalize, and provider would have to continually calibrating based on deployment needs.
> >
> > I tried to answer the question above, and gave the paper another read. That brought me to another point regarding evaluation metrics. The results are primarily reported in terms of AUC, but no threshold-based metrics are provided. While AUC provides a threshold-independent summary, it does not reflect performance at specific operating points relevant to deployment. As a provider, I would prefer TPR at fixed FPR levels (or vice versa) to understand the detector’s performance in realistic deployment scenarios, under realistic constraints, particularly in low-FPR regimes.
> >
> > As for the authors' responses to Q4. I fully appreciate this. Consider adding this to the revised version.
> >
> > At this stage, I maintain my original assessment.

---

> > > ### Author Response · Authors · 2026-04-07
> > >
> > > We thank the reviewer for the continued engagement and the thoughtful follow-up.
> > >
> > > **On the adaptive attack:** While Diaa et al.'s attack is optimized to minimize the global z-score, it achieves this by aggressively removing green tokens from the text, the same fundamental signal that PSS relies on. When the attack degrades the watermark enough to collapse the global z-score to random chance ($\sim$50\%), it has severely degraded the signal that all detectors depend on. PSS maintaining 83.2\% AUC under these conditions demonstrates genuine resilience. A truly PSS-adaptive attack would require white-box access to our feature design and simultaneous optimization against local distributional patterns across all windows and across multiple paraphrase depths, which is a fundamentally stronger threat model than what is standard in the watermarking literature. Beyond the threat model, designing such an attack would also be substantially more computationally costly: the attacker would need to train a paraphraser that simultaneously (i) minimizes our 20-dimensional local feature signature, (ii) destabilizes cross-depth stability across multiple iterative rewrites, and (iii) preserves semantic content and text quality. These three objectives are in tension. Moreover, aggressive feature suppression typically degrades semantic fidelity, making the optimization significantly harder than single-objective global z-score minimization.
> > >
> > > **On generalization across domains:** Table 3 in our paper directly addresses this concern. We trained a single universal classifier on one configuration (Llama-3 + Mistral + PG-19 + D1-D3) and evaluated it on configurations where the LLM, paraphraser, and text domain all changed simultaneously. Specifically, when tested on Qwen2 (different LLM) + Qwen2 (different paraphraser) + WikiText (different domain), the same classifier achieves 90.0\% average AUC *without any retraining or additional calibration*. This is possible because our features capture properties of the watermarking mechanism itself, the keyed hash partition into green/red tokens, which is domain-independent by design. For deployment, a provider can use a single trained classifier across diverse domains and configurations without modification. We note that for highly specialized domains, optional retraining on in-domain data may further improve performance, but it is not a requirement for deployment, a flexibility that, in fact, we view as one of the practical advantages of our approach.
> > >
> > > **On evaluation metrics:** We understand that per ICML's policy, reviewers are not required to review the Appendix, which may have led to this concern. We would like to clarify that threshold-based metrics are already provided in our paper: Table 12 (Appendix A.3) reports TPR at fixed FPR levels across all depths. At 1\% FPR, PSS+Static achieves 84\% TPR at D1 compared to 42\% for global z-score, a 2$\times$ improvement at the operationally relevant threshold. At D8, PSS maintains 64\% TPR versus 24\% for the baseline. We will move these results into the main paper in the revised version to ensure they are immediately visible to readers.
> > >
> > > **On failure case analysis (Q4):** We appreciate the positive feedback and will integrate this into the revised version.
> > >
> > > We believe the above clarifications, combined with new numerical results in the rebuttal including the adaptive attack evaluation, and the TPR@FPR results, demonstrate that the paper's contributions are comprehensive and substantive. We hope the reviewer will consider these in their final assessment.

---

### Official Review · Reviewer_4GXr · 2026-03-12

**Soundness:** 2
**Presentation:** 2
**Significance:** 2
**Originality:** 3
**Overall Recommendation:** 3
**Confidence:** 4

**Summary:**

The paper proposes a custom detector for red-green LLM watermarking schemes, motivated by robustness to paraphrasing and power on short texts. The idea is based on local statistical features and stability dynamics across paraphrased variants; after such feature engineering an XGBoost classifier is trained as the detector. The method is evaluated on AUC across several (dataset, llm generator, llm paraphraser) triples.

**Compliance With Llm Reviewing Policy:**

Affirmed.

**Final Justification:**

I keep my score of 3 post-rebuttal. The authors have pointed out some things that I have missed which was useful but have not addressed several of my other points. I overall feel that the missing experiments and framing issues warrant another revision cycle as such big changes would anyways be too big to make during the rebuttal. The paper is promising and I could see it be accepted eventually, but is in my opinion not ready in the current state.

**Key Questions For Authors:**

Questions:
- Can you explain the claim that paraphrasing breaking up runs of green tokens or relocating them affects the global z-score?
- Can you confirm that D8 experiments only use D8 and D9 for PSS, if so do you have any intuition how it is effective given that much less datapoints are used in this case?
- Can you relate your work to other work on detectors as per my first point above?

**Limitations:**

Limitations are adequately discussed.

**Strengths And Weaknesses:**

The paper studies an important problem (watermark robustness) and motivates the choice of only adapting the detector by retroactive applicability which is reasonable. The idea is to the best of my knowledge novel: it is heavily based on feature engineering but the features are well-motivated and interesting (especially PSS) and some value is demonstrated in experiments. The experiments are relatively comprehensive and include aspects like transferability which is quite important for practical deployment.

However, I find several framing/positioning issues which in my opinion make the paper not ready for publication in the current form:
- Baselines: the paper positions itself mostly against global detectors and robustness-aware schemes but seems to ignore previous attempts that are also focused on only changing the detector while fixing the algorithm, and some of these have explored similar ideas. For example, [1] (one of the first papers on llm watermarking) proposes WinMax and a detector based on run-length differences. [2] studies the idea of using unwatermarked text to aid watermark detection in a Bayesian-like approach. [3,4,5] all study detectors for robustness and/or localization in different ways. These are all prominent works and without proper contextualization the current draft seems to imply a lot more fundamental contribution than it should.
- The post-hoc detector baselines (radar, binoculars) seem somewhat stale. The authors should at least mention [6] which has gained a lot of traction.
- Framing: The paper positions KGW as the only approach to watermarking (especially Introduction) which is misleading to uninformed readers. Later parts of the paper mention more approaches but key ideas different from KGW such as [7,8,9] are not discussed.
- Framing: The authors say (I paraphrase) "watermarking is easy to deploy which is an advantage compared to post-hoc" but post-hoc detectors require no deployment changes so are by definition as easy to deploy. Wording here should be adapted.
- Framing: The authors say that paraphrasing can break long runs of green tokens / relocate them / add local red pockets which affects the global score. But in my understand, these do not affect the global score at all as it is not dependent on position of green tokens. What authors should say, in my opinion, is that paraphrasing keeps substrings of green tokens sometimes intact which implies a localized signal can be extracted that will take that into account when the global score does not due to dillution (also implied in [1]).

On top of this, I have concerns about the evaluation:
- Major: AUC is positioned as the main metric and the only metric studied in the main text. However, the field concensus is that AUC is uninformative for watermarking where no realistic application would use FPR>1%. I appreciate the TPR@FPR results in the appendix but AUC being much more prominent sets a bad example and makes most main paper results fundamentally questionable.
- The quality of paraphrases is not measured, it is unclear if D8 is even legible in this setup.
- The informed paraphrasers [10,11] are ignored. I would not think it is needed to come up with adaptive attacks on PSS in this paper but adaptive paraphrasers have existed for a while and authors only consider naive paraphasing.
- Minor: I find it strange that while short texts are one of the key motivations, main paper results all show only longer texts.
- No code is submitted so it is hard to verify the correctness of the implementation.

Finally, I belive the writing could be improved: the Introduction is unusally long and dense. I suggest the authors separate Page 2 as "Motivation / Requirements" section and reposition "Why global z-score fails" there and ideally place it after the background section which introduces the relevant background. Figures (esp. Fig 1) are very small and not legible without a lot of zooming in. These are minor points in comparison but make the paper extremely unusual in the context of ICML papers.

I am looking forward to the authors response.

[1] "On the Reliability of Watermarks for Large Language Models", Kirchenbauer et al. \
[2] "Scalable watermarking for identifying large language model outputs", Dathathri et al. \
[3] "Efficiently Identifying Watermarked Segments in Mixed-Source Texts", Zhao et al. \
[4] "Detecting Post-generation Edits to Watermarked LLM Outputs via Combinatorial Watermarking", Xie et al. \
[5] "Enhancing LLM Watermark Resilience Against Both Scrubbing and Spoofing Attacks", Shen et al. \
[6] "Technical Report on the Pangram AI-Generated Text Classifier", Emi et al. \
[7] "Robust Distortion-free Watermarks for Language Models", Kuditipudi et al. \
[8] "My AI Safety Lecture for UT Effective Altruism", Aaronson et al. \
[9] "Undetectable Watermarksfor Language Models", Christ et al. \
[10] "Watermark Stealing in Large Language Models", Jovanovic et al. \
[11] "Optimizing Adaptive Attacks against Content Watermarks for Language Models", Diaa et al.\

---

> ### Author Rebuttal · Authors · 2026-03-31
>
> We thank the reviewers for their thoughtful comments and constructive feedback. Below, we respond to the major points raised and would welcome any additional questions or suggestions the reviewers may have.
>
> Q1: There seems to be a misunderstanding. What we meant was that paraphrasing can break up runs or relocate green tokens locally. The Reviewer is correct on the fact that the global z-score depends only on $\sum b_t$ (total green-token count), not on where green tokens appear in the sequence. The accurate characterization is: paraphrasing replaces some green tokens with non-green tokens. This replacement is spatially non-uniform, i.e. some regions retain strong watermark signals while others are heavily edited. The global z-score averages over the full sequence, so concentrated evidence in surviving regions gets diluted by heavily-edited regions. A local detector, on the other hand, recovers this heterogeneous signal by identifying windows where watermark evidence concentrates, which is the core motivation for our approach.
>
> Q2: Yes, confirmed: PSS at D8 uses only D8 and D9 (two data points for the standard deviation). The intuition is that even with two depths, the standard deviation captures a meaningful distinction. For watermarked text, local z-scores at D8 and D9 remain similarly elevated because the watermark bias is anchored to the keyed hash function and persists through paraphrasing, yielding low cross-depth variance.
>
> Q3: First, we implemented WinMax from [1] and evaluated it through our full pipeline. Results on PG-19 at 1,500 tokens (AUC \%):
>
> | Method | D1 | D3 | D5 | D8 |
> |--------|------|------|------|------|
> | Global z-score | 74.2 | 70.0 | 68.0 | 66.8 |
> | WinMax | 82.1 | 76.5 | 74.4 | 72.3 |
> | Static features (20-D) | 88.1 | 83.2 | 80.2 | 78.6 |
> | PSS + Static | **96.1** | **93.9** | **92.6** | **91.2** |
>
> WinMax improves over the global z-score by identifying the strongest surviving local region, but it reduces all local information to a single maximum value. Our approach outperforms WinMax at every depth through two distinct additions: (1) richer 20-D statistics; (2) cross-depth stability. We also thank the Reviewer for pointing to the works [3-5] on detector robustness and localization. Our cross-domain generalization experiments (Tables 2-3) and mixed-paraphrasing evaluation (Figure 5) already address related questions of detector robustness from complementary angles, making these works natural reference points for positioning our contribution. We will incorporate them into our Related Work in the revised version to properly situate PSS within this growing line of detector-side research. We note that [2] is already cited in our paper, though we agree their Bayesian detection perspective deserves more explicit discussion alongside our approach.
>
> Additional comments:
>
> (1) Informed/Adaptive paraphrasers: we thank the Reviewer for pointing out to these related references. Note that our threat model explicitly assumes a black-box paraphraser without access to the watermark key, which reflects the most common real-world deployment scenario and is clearly stated in Section 3.2. That said, to stress-test PSS beyond this threat model, and to further address the Reviewer's comment, we evaluated against the DPO-optimized adaptive attack from [11], which fine-tunes a paraphraser to specifically minimize the global z-score. Under this attack, the global z-score collapses to near random chance (52.1\% at D1, 53.6\% at D2), confirming it is effectively defeated. In contrast, PSS+Static maintains 83.2\% at D1 and 80.6\% at D2, a 30+ percentage point advantage over the global z-score under the same attack.
>
> (2) Quality of paraphrases: This issue is already addressed in the Appendix: Table 13 (Appendix A.3) reports BERT similarity scores at every depth, showing progressive degradation from 0.92 at D1 to 0.68 at D8. The realistic attack range is D1-D3 where semantic similarity remains high ($>$0.85) and PSS+Static achieves 94-96\% AUC. At D8, the text is substantially degraded (BERT similarity 0.68), yet PSS still maintains 91.2\% AUC-meaning an attacker must severely compromise text quality while still being detected at over 91\%, highlighting the fundamental difficulty of evading our approach.
>
> (3) Code: We would like to respectfully note that our anonymous code repository is linked in the footnote on page 6 of the manuscript. The repository contains the complete implementation, including all Python scripts for watermarking, paraphrasing, detection, and classification, structured YAML configuration files with exact hyperparameters, paraphrasing prompts, and decoding settings for all models, as well as a README with setup and reproduction instructions.
>
> (4) Figure 1: We will redesign Figure 1 to span the full page width for improved legibility in the revised version.
>
> (5) References: We will include a detailed discussion on the additional refs provided by the Reviewer in the revised Intro.

---

> > ### Author Rebuttal · Reviewer_4GXr · 2026-04-04
> >
> > Thank you for the rebuttal and for pointing out the code and the paraphraser quality evaluation which I indeed missed. While the additional experiment looks promising, the framing, metric choice and overal positioning of the paper still in my opinion require more work to be ready. I would suggest the authors include the adaptive attack experiments and the comparison to prior work on detectors as more central in the next revision, as also sugested by other reviewers.

---

> > > ### Author Response · Authors · 2026-04-07
> > >
> > > We thank the reviewer for the constructive engagement throughout this process and for acknowledging the additional experiments, code repository, and paraphrase quality evaluation.
> > >
> > > **On metric choice:** We would like to emphasize on the fact that to be comprehensive and transparent in our evaluation, we report both AUC and TPR@FPR throughout the paper. AUC is a threshold-independent metric widely adopted in the watermarking and AI-text detection literature, including SynthID-Text (Dathathri et al., 2024), RADAR (Hu et al., 2023), Binoculars (Hans et al., 2024), DeepTextMark (Munyer et al., 2024), and FreqMark (Xu et al., 2024). Importantly, we do not rely on AUC alone. Table 12 (Appendix A.3) reports TPR@FPR=1\% across all depths, and our conclusions are fully reinforced under this stricter metric: PSS+Static achieves 84\% TPR at D1 vs. 42\% for global z-score (a 2$\times$ improvement), and maintains 64\% vs. 24\% at D8. The same progressive improvement pattern holds under both metrics, confirming that our findings are robust to metric choice. Due to space constraints in the original submission, TPR@FPR results were placed in the appendix; in the revised version, we will restructure the main paper to include both AUC and TPR@FPR=1\% as primary metrics, ensuring both are visible without requiring appendix access.
> > >
> > > **On framing and positioning:** We agree that the WinMax comparison and adaptive attack evaluation should be presented more centrally in the paper, and we commit to restructuring the revised version accordingly. We want to emphasize that all experimental contributions are comprehensive and substantive: (1) PSS+Static outperforms WinMax by 14-19 percentage points in AUC at every depth; (2) under the Diaa et al. adaptive attack, PSS maintains 83.2\% AUC at D1 while the global z-score collapses to $\sim$50\%, a 30+ percentage point advantage. These are definitive results, not preliminary findings. The remaining work, integrating these as core results in the main text and expanding the discussion of related literature, is presentational in nature, and we will fully address it in the revised version to resolve the reviewer's concerns.

---

### Official Review · Reviewer_e72L · 2026-03-13

**Soundness:** 2
**Presentation:** 2
**Significance:** 2
**Originality:** 3
**Overall Recommendation:** 3
**Confidence:** 5

**Summary:**

To address the performance degradation of existing detectors in scenarios involving multiple rewrites and short texts, this paper proposed a watermark detection method specifically for text generated by large language models, named the Pattern Stability Score (PSS). PSS combines global and local z-score features with high-order statistics of run-length patterns, enriched by autocorrelation signals and stability scores based on the depth of rewriting, thereby capturing the statistical anomalies of watermarks and invariant patterns during the rewriting process. The experimental results show that compared to traditional z-score threshold methods and several advanced deep learning methods, PSS achieves an improvement of over 10-15 percentage points in the detection AUC across different text lengths.

**Compliance With Llm Reviewing Policy:**

Affirmed.

**Final Justification:**

Overall, I find the paper technically sound and potentially useful, with clear empirical promise, but still somewhat limited in clarity and in how convincingly it establishes its conceptual novelty and practical setting. My final recommendation reflects this balanced view: the rebuttal meaningfully strengthened the submission and improved my assessment, but some important concerns remain.

**Key Questions For Authors:**

See weaknesses.

**Limitations:**

Yes

**Strengths And Weaknesses:**

Strengths:
1. A z-score calculation method based on local sliding windows was designed, providing a novel technical idea for text watermark detection.
2. The performance under different rewriting levels, cross-domain migration scenarios, and various token quantities was investigated.
Weaknesses:
1. The logic of the introduction section needs to be strengthened. This paper has overly elaborated on the advantages of detector optimization, failing to closely focus on the core issue of this article, which results in an unclear main thread.
2. The implementation details of the rewriting attack are insufficiently described. Although the rewriting levels (D1–D8) are defined in the text, the corresponding modification operations for each level are not specified, which affects the reproducibility of the experiment. Moreover, the possibility that the modification process itself may introduce new watermarks and thereby interfere with the detection effect has not been considered.
3. Rewriting attacks in the watermarking field have been studied extensively (e.g., [1]), but the authors did not explore the effectiveness of their detection method against existing rewriting attack approaches. Instead, they independently designed a rewriting attack, which makes the experimental comparison insufficient and makes it difficult to identify the actual advantages of the method presented in this paper.
4. Although this paper focuses on the task of text watermark detection, in the experimental comparison, the selected baseline methods mainly come from the AIGC detection field rather than the current mainstream watermark detection technologies. This weakens the targeted nature of the comparison and the persuasiveness of the conclusion. Moreover, the authors did not further verify the applicability or effectiveness of their proposed method in other watermark schemes, which makes the completeness of the experimental evaluation somewhat lacking.
5. The practicality of the proposed detection protocol is unclear. Although the paper states that the detector receives a single text at test time, the PSS computation requires generating additional paraphrases from the test input across subsequent depths. This departs from the standard watermark detection setting and introduces extra assumptions, computational overhead, and dependence on the chosen paraphraser. The paper should clarify whether this is a realistic deployment scenario and how sensitive the results are to the detector-side paraphrasing process.
6. The methodological novelty of the paper still needs to be better justified. The main contribution appears to lie more in feature engineering and classifier combination, rather than in a fundamentally new watermark detection framework.
7. Some important experimental details are not sufficiently specified, which affects the reproducibility of the work. In particular, several implementation choices and parameter settings remain unclear.
8. The ablation study is not yet sufficiently convincing. The current experiments do not fully clarify the individual contribution of each component, making it difficult to determine which parts are truly responsible for the performance gains.

---

> ### Author Rebuttal · Authors · 2026-03-31
>
> We thank the reviewer for their thoughtful and constructive feedback. We address major comments below; while many raise valuable points, some reflect misunderstandings that we clarify in our responses.
>
> Strength 1-2: thank you for the positive feedback.
>
> Weakness 3: We will move the extended detector-centric justification into a dedicated subsection, keeping the Intro focused on the problem statement, key idea, and contributions.
>
> W 4-5: First, the Reviewer's reference [1] is missing so we could not investigate the additional rewriting attack. Second, our modification operations are available in the structured configuration files of our anonymous code repository (footnote, page~6). Third, following this feedback, we evaluated against DPO-optimized adaptive attack from [1], and results are provided in response to Q3 of Reviewer rBos.
>
> W 6: In terms of selecting competing methods, we included both AIGC detectors (DeepTextMark, Binoculars, RADAR) and statistical-based ones (global z-score, etc.). To further address the Reviewer's comment, we included an additional competing method (WinMax [2]) and its results are summarized in response to Q3 of Reviewer 4GXr. This shows that our selected benchmark detectors cover a wide range of methods.
>
> W 7: This comment is already addressed in response to Q1 of Reviewer 3RRv.
>
> W 8: We respectfully but firmly disagree with this characterization, which we believe represents a significant oversimplification of our findings and does not fully reflect the conceptual contributions of the work. Our work makes two significant conceptual contributions:
>
> First, we identify a structural property of watermark signals under paraphrasing: the degradation is spatially non-uniform. Paraphrasing does not remove watermark evidence evenly across the text, it creates a mosaic of heavily-edited and well-preserved regions. While prior work such as WinMax has exploited locality by selecting the strongest window, no prior approach has systematically characterized this heterogeneity through multi-dimensional statistical summaries capturing distributional shape, spatial dependencies, and run-length structure simultaneously.
>
> Second, the PSS functional introduces a fundamentally new detection dimension: temporal stability across rewrites. Prior detectors - including WinMax, global z-score, and deep learning approaches, all analyze a single version of the text. PSS is the first to measure how local watermark evidence persists across successive transformations. This is a conceptually distinct signal: rather than asking "does this text look watermarked?'' we ask "does the watermark evidence in this text behave consistently under perturbation?'' This stability perspective has no precedent in the watermark detection literature.
>
> The practical consequence of these insights, that interpretable statistical features grounded in spatial and temporal structure dramatically outperform both simple local detectors and black-box neural architectures, is itself a finding with implications for future detector design.
>
> W 9: We respectfully but firmly note that all relevant experimental details are already specified in the manuscript, with additional implementation and evaluation details documented in the Appendix for completeness. Further, some additional details including exact paraphrasing prompts and decoding hyper-parameters are provided in our anonymous code repository (footnote, page 6) as structured YAML configuration files.
>
> W 10: Following this feedback, we conducted a finer-grained feature-group ablation within the 20-D static features to isolate each component's contribution. Results on PG-19 at 1,500 tokens (AUC \%):
>
>
>
> | Feature Set | D1 | D3 | D5 | D8 |
> |-------------|------|------|------|------|
> | Global z-score (baseline) | 74.2 | 70.0 | 68.0 | 66.8 |
> | Z-score moments (6-D) | 83.2 | 76.5 | 74.1 | 72.0 |
> | + Autocorrelations (8-D) | 85.0 | 79.3 | 76.0 | 74.4 |
> | + Run-length stats (14-D) | 87.3 | 83.2 | 79.8 | 77.2 |
> | + Run-frequency (full 20-D) | 88.1 | 83.2 | 80.2 | 78.6 |
> | + PSS (PSS+Static) | **96.1** | **93.9** | **92.6** | **91.2** |
>
>
> Z-score moments carry strong individual contribution, establishing the baseline for local detection. Autocorrelations add temporal dependency information that helps at all depths. Run-length and run-frequency statistics capture the structural signature of greenlist biasing, with their contribution becoming more pronounced at deeper paraphrase depths where the z-score signal alone weakens. PSS provides a large single performance jump by adding the cross-depth stability dimension. This empirical investigation confirms that each component addresses a distinct aspect of the watermark signal, and none are redundant.
>
> [1] "Optimizing Adaptive Attacks against Content Watermarks for Language Models", Diaa et al.
>
> [2] "On the Reliability of Watermarks for Large Language Models", Kirchenbauer et al.

---

> > ### Author Rebuttal · Reviewer_e72L · 2026-04-04
> >
> > Thank you for the detailed rebuttal and for the additional experiments. Your responses helped clarify several points, especially the finer-grained ablation, the added comparison with WinMax, and the additional evaluation against adaptive attacks. These additions improve the paper and partially address my earlier concerns, so I will raise my score by one point.
> >
> > I also want to apologize for an omission in my original review: in Weakness 5, I referred to prior rewriting-attack work as “[1]” but did not provide the full citation. The reference I intended was: [1] Cheng, Yixin, et al. “Revealing Weaknesses in Text Watermarking through Self-Information Rewrite Attacks.” arXiv preprint arXiv:2505.05190 (2025). Sorry for the missing citation.
> >
> > That said, I still believe some concerns remain, particularly regarding the practical deployment assumptions of detector-side paraphrasing and the need to more clearly position the methodological novelty relative to strong feature engineering. Still, I appreciate the effort taken in the rebuttal, and I acknowledge that the revision meaningfully strengthens the work.

---

> > > ### Author Response · Authors · 2026-04-07
> > >
> > > We sincerely thank the reviewer for the constructive engagement, for adjusting the score, and for providing the full SIRA citation (Cheng et al., 2025), which we will cite in the revised version. Regarding the two remaining concerns:
> > >
> > > **On deployment practicality:** Our framework provides two complementary deployment paths that practitioners can choose between based on their requirements. The first is the static features (20-D) tier, which requires no test-time paraphrasing, runs in sub-second time, and is computationally as lightweight as the global z-score detector. Despite this minimal cost, static features alone achieve 88.1\% AUC at D1 and 78.6\% AUC at D8 on PG-19 (1,500 tokens), substantially outperforming both the global z-score (74.2\% $\to$ 66.8\%) and WinMax [1] (82.1\% $\to$ 72.3\%) at every depth. This makes static features a strong, practical baseline for latency-sensitive deployments where paraphrasing is not feasible. The second tier is the full PSS+Static pipeline, which requires paraphrase generation and adds 8-13 percentage points of AUC across depths (reaching 96.1\% at D1 and 91.2\% at D8). This tier is intended for high-stakes applications such as legal evidence, academic integrity verification, and content provenance, where the additional computational cost is justified by the substantially reduced miss rate. The paraphrasing cost is therefore a deliberate accuracy-latency tradeoff that practitioners control, not an architectural limitation.
> > >
> > > **On novelty positioning:** Besides our previous response, we would like to clarify our contribution from two complementary angles.
> > >
> > > **(A) Empirical evidence:** The strength of our methodological contribution is demonstrated by the fact that both PSS+Static and even our static features alone substantially outperform every baseline we tested including global z-score, WinMax, and three deep learning post-hoc detectors (DeepTextMark, Binoculars, RADAR, all in the 41-44\% AUC range at D8). Under the Diaa et al. adaptive attack [2] added during rebuttal, PSS+Static maintains 83.2\% AUC at D1 while the global z-score collapses to $\sim$50\%. This is a 30+ percentage point advantage under a strong adaptive attack available in the watermarking literature. These results have direct real-world implications: PSS makes it possible to reliably verify watermarked content in deployment scenarios, legal disputes, academic integrity, content authenticity, where many existing detectors fail.
> > >
> > > **(B) Conceptual interpretation:** Our work introduces a fundamentally different way of thinking about watermark detection. The global z-score asks "how much watermark signal is present in this text?''; deep learning post-hoc detectors ask "does this text statistically resemble AI-generated text?''; WinMax asks "what is the strongest local concentration of signal?'' Our PSS framework asks a qualitatively different question: "how does the watermark evidence in this text behave under perturbation?'' To the best of our knowledge, this stability-based perspective has no precedent in the watermark detection literature and turns paraphrasing, traditionally treated as an attack to be defended against, into a source of discriminative signal. We believe this conceptual advancement, validated by consistent and substantial performance gains across all of our numerical experiments, constitutes a meaningful contribution beyond standard feature engineering.
> > >
> > > We end by noting that with the finer-grained ablation, WinMax comparison, Diaa et al. adaptive attack evaluation, and our two-tier deployment design, we believe the paper now addresses the reviewer's concerns with both concrete empirical evidence and a clear conceptual positioning of our contribution.
> > >
> > > [1] "On the Reliability of Watermarks for Large Language Models", Kirchenbauer et al.
> > >
> > > [2] "Optimizing Adaptive Attacks against Content Watermarks for Language Models", Diaa et al.

---

### Official Review · Reviewer_3RRv · 2026-03-13

**Soundness:** 3
**Presentation:** 3
**Significance:** 3
**Originality:** 2
**Overall Recommendation:** 4
**Confidence:** 4

**Summary:**

Watermarking methods might be unrobust under paraphrasing.
The paper, instead of trying to resist paraphrasing, changes the detector to exploit signals that paraphrasing preserves.
They propose applying local rolling window statistics as the feature across multiple rounds of paraphrasing.
The results show that the results are still high after 8 rounds of paraphrasing, while other baselines significantly degraded.

**Compliance With Llm Reviewing Policy:**

Affirmed.

**Final Justification:**

The paper overall is good in soundness, presentation, and significance. Most of my concerns are addressed, and the rest I see as a limitation that is acceptable from my perspective. I maintain my positive assessment (score=4) and recommend the paper to be accepted.

**Key Questions For Authors:**

1. The methods require additional computation, i.e., at least one time rephrasing. This is a weakness of the method. I wonder if cheaper perturbation methods, e.g., smaller paraphraser models, or deterministic perturbation methods, could replace the current large paraphraser models in the trained time and test time, but still have robustness under large paraphrasers.
2. Do the other baseline methods also train on 8 paraphrasing versions of the training corpus? Otherwise, it is not a fair comparison since your methods have extra augmented data.
3. I believe the efficacy of PSS has not been fully accounted for. If the assumption is that watermarks -> low PSS, while human-written content -> high PSS, then we ought to test whether PSS maintains consistent results solely for LLM paraphrases (i.e., robust), whilst human rewrites tend to lower confidence in machine-generated content. Only then does the entire logic form a closed loop.
I see Table 14 in the appendix, which seems to support this, but needs more clarification.

**Limitations:**

Yes.

**Strengths And Weaknesses:**

Strength:
1. The paper has a clear presentation of the methods.
2. The results show better robustness of the methods. And it covers testing generalization performance.
3. The method is original in this specific domain.

Weakness:
1. The introduction parts are long. Consider summarizing it and moving fake paragraphs to separate sections.
2. The methods require additional computation, i.e., at least one time rephrasing.
3. The choice of metric of 20-D features lacks motivation and explanation of the reason.

---

> ### Author Rebuttal · Authors · 2026-03-31
>
> We thank the reviewers for their thoughtful comments and constructive feedback. Below, we respond to the major points raised and would welcome any additional questions or suggestions the reviewers may have.
>
> Q1: Our framework was intentionally designed to be modular with flexible deployment tiers, as detailed in Tables 15-16 (Appendix). To clarify the computational breakdown: PSS detection itself (feature extraction and classification) requires only 0.8-3.2 seconds for 300-1,500 token passages (Table 15). The computational overhead the reviewer refers to comes from paraphrase generation, not from our method. In the static-features-only tier, our detector runs in sub-second time with no paraphrasing required, already outperforming all baselines (83-85\% AUC at D8 vs. 62-66\% for global z-score and 41-44\% for deep learning methods). When higher accuracy is needed, PSS with a single paraphrase adds 6-13 percentage points, pushing performance above 90\% AUC, the paraphrasing cost here is a deliberate accuracy-latency tradeoff, not an inherent limitation of PSS.
>
> Regarding cheaper alternatives: PSS is demonstrably paraphraser-agnostic - we evaluate with three architecturally distinct models (Mistral-7B, Gemma-7B, Qwen2-7B) and alternating mixed-model paraphrasing (Figure 5), with consistent rankings across all configurations. This confirms that the stability signal is a fundamental property of the watermark mechanism, not an artifact of any specific paraphraser, strongly suggesting that lighter-weight alternatives would preserve this signal.
>
> Q2: Fairness is an important aspect of our empirical investigation. While some benchmark methods have different detection paradigms, we took certain measures to ensure fairness across methods: Binoculars is a zero-shot method requiring no training at all. RADAR uses its own adversarial training procedure with built-in data augmentation. DeepTextMark was retrained on our data following the protocol described in the original paper. Each baseline operates under its intended paradigm.
>
> More critically, the fairness question is directly resolved by our static features: the 20-D static features are computed from a single text with no cross-depth information and no paraphrased augmentation, exactly the same input setting as all baselines. These static features achieve 83-85\% AUC at D8, compared to 41-44\% for the deep learning baselines. This 40+ percentage point gap under identical input conditions confirms that the performance advantage stems from our feature design, not from any data augmentation imbalance. PSS then adds a further 12.6 percentage point gain through the cross-depth stability dimension.
>
> Q3: Our paper addresses this in the Appendix. Table 14 (Appendix A.3) evaluates PSS under human-like perturbations: with 20\% manual edits on watermarked text, PSS+Static achieves 91.8\% AUC (only a 3.6\% drop); with 30\% manual edits, 88.9\%; and under mixed-model chain paraphrasing, 90.6\%. If the stability signal were merely an artifact of the LLM paraphrasing process rather than the watermark itself, manual edits would destroy the signal entirely, which they clearly do not. The reason is fundamental: watermarked text carries a statistical bias anchored to the keyed hash function that persists regardless of whether the text is rewritten by an LLM or edited by a human. Human text has no such anchor, so its local z-scores fluctuate across variants without any consistent pattern. PSS captures exactly this asymmetry, and the consistently high AUC across all configurations (91-96\% across depths, datasets, LLMs, and paraphrasers in Tables 1-11) confirms robust class separation under diverse perturbation types.
>
> Weakness 1: We will restructure the Intro accordingly in the revised version.
>
> Weakness 3: Section 3.3 of our paper describes the motivation for each feature group: (i) z-score summary statistics (6 features) capture the distribution of local watermark strength across windows; (ii) lag-1 and lag-2 autocorrelations (2 features) capture short-range spatial dependencies that distinguish the coherent local structure of watermarked text from the near-zero autocorrelation of human text; and (iii) run-length and run-frequency statistics (12 features) directly capture the structural signature of greenlist biasing, consecutive green-token runs and their frequency, which paraphrasing fragments in predictable ways.
>
> To provide further empirical justification, we conducted a feature-group ablation. At D8 on PG-19 (1,500 tokens): global z-score achieves 66.8\%, z-score moments alone (6-D) reach 72.0\%, adding autocorrelations (8-D) improves to 74.4\%, adding run-length (14-D) reaches 77.2\%, the full 20-D achieves 78.6\%, and incorporating PSS yields 91.2\%. Every feature group contributes measurable gains, and PSS provides the largest single jump of 12.6 percentage points, confirming that each component captures a distinct and non-redundant aspect of the watermark signal.

---

> > ### Author Rebuttal · Reviewer_3RRv · 2026-04-02
> >
> > Thank the authors for their response. I see the contribution of this paper and the limitations are acceptable under the scope. I maintain my positive assessment.

---

> > > ### Author Response · Authors · 2026-04-07
> > >
> > > We sincerely thank the reviewer for the positive assessment, for confirming that our responses adequately addressed the concerns, and for the thoughtful and constructive feedback throughout the review process. We are grateful for the reviewer's recognition of the paper's contribution and the careful engagement with our work.
> > >
> > > We hope our responses to the reviewer's questions, particularly the deployment tier analysis (Q1), the fairness clarification with the static features comparison (Q2), and the closed-loop validation evidence from Table 14 with manual edits and mixed-model paraphrasing (Q3), together with the additional experiments conducted during the rebuttal (WinMax baseline [1], finer-grained feature ablation, and Diaa et al. adaptive attack evaluation [2]), demonstrate the robustness and completeness of our contribution. We deeply appreciate the reviewer's continued support of this work.
> > >
> > > [1] "On the Reliability of Watermarks for Large Language Models", Kirchenbauer et al.
> > >
> > > [2] "Optimizing Adaptive Attacks against Content Watermarks for Language Models", Diaa et al.

---

### Decision · Program_Chairs · 2026-04-30

**Decision:**

Accept (regular)

**Comment:**

The paper proposes a watermark detection method PSS, tailored for paraphrasing attack. PSS combines global and local z-score features with high-order statistics of run-length patterns, enriched by autocorrelation signals and stability scores based on the depth of rewriting, thereby capturing the statistical anomalies of watermarks and invariant patterns during the rewriting process.

Reviewers like the following strengths of the paper:
1. The paper is motivated well. The method tries to address the detection problem from a different angle.
2. The experimental results show better robustness of the methods.

Reviewers also point out the following weakness:
1. The introduction section could be revised and condensed.
2. More implementation details could be added.
3. Individual component's contribution could be studied and added.
4. Missing comparison with several baselines.